# State-dependent neural representations of muscle synergies in the spinal cord revealed by optogenetic stimulation

Borong He[1] , Paola Salmas[1], Jing Zhang[2], Xiaojie Duan[3] and Vincent C. K. Cheung[1]

[1] *School of Biomedical Sciences, Gerald Choa Neuroscience Institute, The Chinese University of Hong Kong, Shatin, NT, Hong Kong, China*
[2] *The James Franck Institute, The University of Chicago, Chicago, Illinois, USA*
[3] *Department of Biomedical Engineering, College of Future Technology, Peking University, Beijing, China*

Handling Editors: Richard Carson & Matthew Fogarty

The peer review history is available in the Supporting information section of this article (https://doi.org/10.1113/JP288073#support-information-section).

**The Journal of Physiology**

**Abstract figure legend** Spike trains and muscle activities elicited by optogenetic stimulation revealed state-dependent encoding of muscle synergies by upstream spinal interneurons.

B. He and P. Salmas contributed equally to this work.

**Abstract** The central nervous system controls movement by combining neuromotor modules, known as muscle synergies. Previous studies suggest that spinal premotor interneurons (PreM-INs) contribute to the encoding of stable muscle synergies for voluntary movement. But descending and sensory inputs also influence motor outputs through the spinal interneuronal network, which may be configured by its inputs to encode different sets of muscle synergies depending on the network state, thereby recruiting different selections of synergies. Here we tested this possibility of state-dependent synergy encoding by examining the muscle synergies represented by the same upstream spinal interneurons under different activity states induced by various optogenetic stimulation patterns. Lumbosacral spinal units and electromyographic (EMG) activities of hindlimb muscles were simultaneously recorded from anaesthetized Thy1-ChR2 mice as the spinal cord was stimulated by one or two optic fibres at different intensities. The synergy encoded by each unit was revealed as a 'muscle field' derived from spike-triggered averages of EMG, whereas the entire muscle synergy set was factorized from the EMG. We found that although the muscle synergy set remained stable across stimulation conditions, the muscle fields of the same units were matched to different synergies within the set in different states. Thus the interneurons may flexibly adjust their connectivity with the motoneurons of the muscles as descending and sensory afferents impose different states on the spinal network. State-dependent encoding of muscle synergies may allow different synergies to be selected for producing stable movement in an ever-changing workspace environment.

(Received 7 November 2024; accepted after revision 3 July 2025; first published online 28 July 2025)

**Corresponding author** Vincent C. K. Cheung: School of Biomedical Sciences, Gerald Choa Neuroscience Institute, The Chinese University of Hong, Shatin, NT, Hong Kong, China. Email: vckc@cuhk.edu.hk

**Key points**

- Muscle synergies for locomotion can be represented by spinal interneurons, as revealed by the interneurons' muscle fields derived from spike-triggered averages of EMG.
- The muscle field of a single spinal interneuron may vary under different stimulation conditions, as demonstrated by optogenetic stimulation.
- Encoding of muscle synergies is dependent on the state of spinal activities, thus facilitating the selection of appropriate synergies in different dynamic environments.

## Introduction

One main function of the CNS is to co-ordinate a large number of neuromusculoskeletal elements, including the thousands of motor units within the hundreds of muscles, to generate fast, flexible and ecologically relevant motor behaviours (Graziano, 2006). Previous studies have suggested that the CNS organizes complex motor activities by grouping small sets of muscles together, through neural networks in the spinal cord, into basic motor modules known as muscle synergies. The synergies are then selected and combined together to generate different

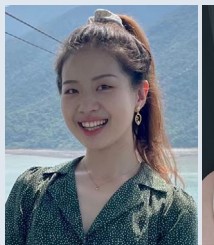
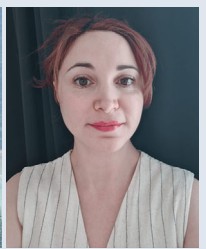

**Borong He** holds a bachelor's in electronic information engineering from Xidian University and a master's in electrical engineering from the National University of Singapore. She is currently a PhD candidate in biomedical sciences at The Chinese University of Hong Kong, focusing on how the spinal cord co-ordinates motor modules for movement. By integrating optogenetic stimulation with multilevel electrophysiological recordings she seeks to uncover mechanisms underlying spinal motor control and inform strategies for restoring motor function after injury. **Dr Salmas's** background is in brain processing, with focuses on motor control, consciousness and perception. She earned degrees in psychology and cognitive psychology and information processing, studying the neural basis of consciousness and neurocognitive simulations. Her PhD in robotics, neuroscience and nanotechnologies centred on primate motor control for brain–machine interfaces and speech perception in humans. At The Chinese University of Hong Kong, she investigated neural motor control in rodents using electrophysiology and optogenetics.

movements (Bizzi & Cheung, 2013; Cheung & Seki, 2021; Tresch et al., 1999).

In the literature many studies have defined muscle synergies as time-invariant vectors that represent the muscles' coactivation patterns (Saltiel et al., 2001). Synergies are recruited by the CNS through their activation coefficients, which are time-varying profiles that specify the temporal variation of how the muscle synergies are scaled in the movement (d'Avella & Bizzi, 2005). Even though muscle synergies have been identified from EMG data of different motor behaviours, such as walking and running (Cappellini et al., 2006), in healthy human subjects and patients with brain damage (Cheung et al., 2012), and across developmental and ageing stages (Baggen et al., 2020; del Vecchio et al., 2020; Guo et al., 2022; Sylos-Labini et al., 2020), the detailed neuronal mechanisms that govern the assembly of selected muscles into muscle synergies and the synergies' selection and combination for movement execution are yet to be explained (Cheung & Seki, 2021).

Previous evidence has suggested that different sets of spinal premotor interneurons could produce synergistic motor outputs by activating the motoneuronal pools of groups of muscles simultaneously. This has been demonstrated in the spinalized frog (Bizzi et al., 1991; Hart & Giszter, 2010; Mussa-Ivaldi et al., 1994), mouse (Levine et al., 2014), rat (Tresch et al., 1999) and monkey (Yaron et al., 2020). To identify the exact contributions of different interneuronal populations to motor behaviours, the interneuron's post-spike facilitation effects on the motoneuronal pools were often calculated using spike-triggered average (SpTA) of EMG (Fetz et al., 2002). Premotor interneurons in the spinal ventral horn (at depth of about 700 μm from dorsal spinal surface) of frogs exhibited post-spike facilitations with significant effects on multiple muscles at long latency, and these neurons are thought to be involved in the neuronal representation of muscle synergies (Hart & Giszter, 2010). In a more recent study Takei et al. (Takei et al., 2017) recorded activities from cervical spinal last-order premotor interneurons (PreM-INs) and hand and arm muscles of two macaque monkeys during a precision grip task. They demonstrated that the 'muscle fields' of the recorded PreM-INs, derived from the neurons' magnitudes of post-spike facilitation on the muscles, were distributed as clusters that matched well with the muscle synergies for the task. The neural population activity of the muscle field clusters also corresponded well to the temporal profiles of the synergies. These data argue that the PreM-INs participate in the structuring of the muscle synergies employed during voluntary hand movements. Also, intracortical micro-stimulation studies in the monkey's motor cortex have shown that behavioural synergies could be elicited by motor cortical stimulations (Overduin et al., 2012), thereby suggesting that the synergy-encoding PreM-INs are recruited by descending corticospinal neurons (Rathelot & Strick, 2006).

Other studies in the literature have demonstrated the existence of neuromotor modules by directly stimulating the spinal grey matter (Bizzi et al., 1991). For example, researchers (Caggiano et al., 2016) compared the optogenetically evoked motor co-ordinative patterns from two strains of transgenic mouse, Chat-ChR2 (with channelrhodopsin expressed in motoneurons) and Thy1-ChR2 (with channelrhodopsin expressed in Thy1-positive spinal interneurons). They found that the isometric force patterns at the ankle elicited from the anaesthetized Thy1-ChR2 mice were more complex than those from the Chat-ChR2 mice. It has been further shown in a subsequent study (Salmas & Cheung, 2023) that interneurons through the entire length of the lumbosacral spinal cord contained a topographic organization of different co-ordinative force modules with spatially overlapping representations. These data suggest that the spinal interneuronal networks, beyond just the PreM-INs, may underpin the emergence of complex motor co-ordinative patterns necessary for multiple behaviours.

While the activities of a specific group of neurons may encode the muscle synergies, it is also clear that inputs from the peripheral receptors and supraspinal pathways can strongly influence motor outputs via spinal interneurons (Pearson, 1981). Recent findings suggest that the spinal network encodes a set of default muscle synergies (Desrochers et al., 2019) whose structures could be fine-tuned by descending and sensory inputs. These inputs engage upstream spinal interneurons, dynamically shaping the default synergies through complex synaptic connections and firing properties (Yang et al., 2019) to accommodate various behaviours (Cheung, Zheng, et al., 2020). Also, to adapt to the constantly changing environmental demands, the same spinal neuronal network may even flexibly encode varied muscle synergies depending on the state of its activity imposed by feedback and descending modulation. Indeed previous data have shown that different reflex-driven muscle patterns can be mediated by the same afferent pathways in the spinal cord depending on the spinal state (Pearson & Collins, 1993). However whether state-dependent encoding of muscle synergies is implemented in the spinal cord, and, if so, what the neural mechanisms of its implementation are, has remained unclear.

In this study we aim to elucidate the role of upstream spinal interneurons, beyond spinal PreM-INs, in generating hindlimb muscle co-ordination patterns by examining the relationship between these interneurons and motoneuronal pools under different spinal states induced by optogenetic stimulation. We investigate how neural commands selectively recruit muscle synergies as neuromotor modules to produce robust movements. Using optogenetics, as in (Caggiano et al., 2016),

we selectively stimulated Thy1-positive inter-neurons while recording spinal interneuron activities and EMGs from multiple hindlimb muscles. Our findings demonstrate that hindlimb muscle synergies derived from EMG could be retrieved from interneuronal muscle fields. Additionally different spinal stimulation conditions altered the muscle fields of individual units in distinct patterns; yet these altered muscle fields could still be matched to a different set of muscle synergies. This suggests that the CNS flexibly modulates spinal encoding of muscle co-ordination patterns in a state-dependent manner to produce stable motor behaviours.

## Materials and methods

### Preparation and animal surgeries

**Animals and ethics.** Nine transgenic mice (20–30 g, male, Thy1-COP4/EYFP; The Jackson Laboratory), with light-gated channelrhodopsin-2 (ChR2) expressed primarily in excitatory interneurons, were used (Caggiano et al., 2016; Wang et al., 2007). Each experiment consisted of a session for implantation of intramuscular EMG electrodes in hindlimb muscles followed by a second session on the next day for both neural electrode implantation and the spinal stimulation experiment. Before each session the mouse was anaesthetized first with isoflurane (1.5%–2%; mixed with 0.8 L/min oxygen), followed by an intraperitoneal injection of ketamine/xylazine (100/10 mg/kg) before implantation of intramuscular EMG electrodes and laminectomy. On the first day of surgery, as part of postoperative care, buprenorphine (0.05 mg/kg) was administered sub-cutaneously to alleviate pain. During the experiment, to ensure that the animal remained under deep anaesthesia, we assessed the animal every 20 min by pinching the tail and foot with a pair of tweezers and by checking the blink reflex. If the animal exhibited any reflexive motor responses, an additional dose of anaesthetic (half of the initial dose) was delivered, and then a 15–20 min break was taken before continuing. At the end of the experiment, while still under deep anaesthesia, the animal was perfused transcardially as described in the section 'Histology'. Throughout the entire experiment each animal was anaesthetized once before the terminal procedure. Although the use of anaesthesia ensures stable stimulations and recordings, we acknowledge this as a limitation of our study. Before and after experiments, mice were housed and fed (rodent diet and clean water *ad libitum*) in a dedicated room within the animal care facility of the School of Biomedical Sciences at The Chinese University of Hong Kong. All procedures were approved by the Animal Experimentation Ethics Committee of The Chinese University of Hong Kong (CUHK) (protocol 20-282-MIS) before experimentations. Animal care was provided in accordance with CUHK guidelines.

**Intramuscular EMG electrode preparation and implantation.** Six pairs of intramuscular EMG electrodes were prepared the day before the implantation session following a procedure modified from Pearson et al. (Pearson et al., 2005). Each pair of electrodes was fabricated by twisting and knotting two stainless steel wires (diameter of 0.001'; length of 8–10 cm; A-M Systems catalogue no. 793200). The insulation was removed at 2 mm from the knot for one wire and 3 mm for the other to detect changes in electrical signals along the muscle fibres. Buprenorphine (0.05 mg/kg) was administered intraperitoneally 10 min before electrode implantation. Skin incision was then made above the muscles to be implanted, including tibias anterior (TA), vastus lateralis (VL), vastus medialis (VM), bicep femoris (BF), gluteus maximus (GM) and the gracilis group (GC) on the animal's left side. The anatomical locations of the muscles were identified using atlases of the mouse hindlimb (Charles et al., 2016). The electrodes were inserted into the belly of the muscle with a leading needle. After that individual pin connectors (A-M Systems catalogue no. 520100) were crimped to the ends of the electrode wires that emerged from the skin incisions. The incisions were then sutured with surgical nylon wires, and the electrode wires were taped together.

**Carbon nanotube fibre electrodes.** The bulk size and rigid mechanical properties of traditional electrode implants for neuronal tissues can lead to tissue injury and sustained inflammatory response, which contribute to potential failure of neural signal recording (Jeong et al., 2015; Salatino et al., 2017). In contrast, soft and flexible carbon nanotube fibre (CNTF) electrodes are ideal candidates for implantable depth electrodes due to their improved electrochemical performance, mechanical compliance and tensile strength. Moreover, because of their flexibility, CNTF electrodes do not require any holder for mechanical support at the recording sites; thus the use of CNTF in our setting would leave the exposed spinal cord surface free of obstacles. This in turn would allow us to comprehensively stimulate the lumbosacral spinal cord by moving the laser in small incremental steps along the entire exposed surface. CNTF electrodes have been demonstrated to be capable of continuously detecting and isolating single neuronal units from the rat brain for up to 4–5 months (Lu et al., 2019) and from the spinal cords of the free-moving rat for 3–4 months without the need of repositioning the electrodes (Liu et al., 2022).

**Carbon nanotube fibre electrode fabrication.** The CNTFs were spun out of spinnable CNTF arrays produced by the chemical vapour deposition method described earlier (Jiang et al., 2011; Liu et al., 2010). Briefly CNTF arrays were grown on $SiO_2/Si$ wafer substrates coated with 1–2 nm thin Fe film, which worked as a catalyst. The growth was processed for 15 min at 750°C at atmospheric pressure, with $C_2H_4$ of 300 sccm (standard cubic centimetres per minute) as the carbon source diluted in $H_2$ of 200 sccm and Ar of 1000 sccm. This resulted in vertically aligned CNTF arrays consisting of double- and triple-walled CNTFs with an average diameter of 6 nm. The fibres were spun with a homemade spinning machine. During spinning, ethanol was rushed onto the fibre surface to densify the fibre and clean the fibre from dust.

Here we fabricated electrodes for intraspinal recording using CNTFs with a diameter of 15 μm and coated with a 2 μm thick layer of Parylene-C film as an insulating layer (Fig. 1B). These fabricated CNTF electrodes show great flexibility, as they can swing freely indoors and have a bending stiffness per width that is two orders of magnitude lower than those in traditional silicon probes (Lee et al., 2005; Liu et al., 2022). This property makes the CNTF electrodes have mechanical properties more similar to those of neural tissues and reduces tissue damage during recording (Tyler, 2012).

**Laminectomy and carbon nanotube fibre electrode implantation.** To expose the lumbosacral vertebrae, the skin from the thoracic to the sacral region was shaved, cleaned with 70% ethanol and incised. The soft tissues and paraspinal muscles were then dissected using sterilized surgical tweezers and scissors to expose the spinous processes and laminae of the T11 to T13 thoracic vertebrae. The exposed vertebrae were secured with vertebral clamps in a stereotaxic apparatus. A laminectomy was performed to remove the spinous processes, laminae and ligamentum flavum of the T11, T12 and T13 vertebrae, bones that house the lumbosacral spinal cord (Harrison et al., 2013).

Once the spinal cord was exposed, tungsten wires were used as shuttles to guide implantation of the soft CNTF electrodes. The electrodes were adhered to the parallel tungsten wires (diameter of 50 μm) using an 8% polyethylene oxide (PEO) solution. After drying, the complex of CNTF electrodes and tungsten wires was inserted into the target region. To prevent the separation of the CNTF electrodes from the shuttles before the electrode tip reached the target depth due to PEO dissolution, the insertion process was always completed within 5 s. The tungsten wires were removed by soaking the implanted spinal surface with standard saline solution for 5–10 min, leaving the CNTF electrodes in place inside the tissues.

## Optogenetic stimulation and electrophysiology recording

**Optogenetic stimulations.** Stimulations were delivered to the exposed spinal cord by a diode-pumped solid-state blue laser (473 nm; NEWDON, Shanghai, China) coupled with two optical fibres with a 200 μm core. In the first part of the experiment, one laser fibre was positioned above the most rostral location of the exposed spinal

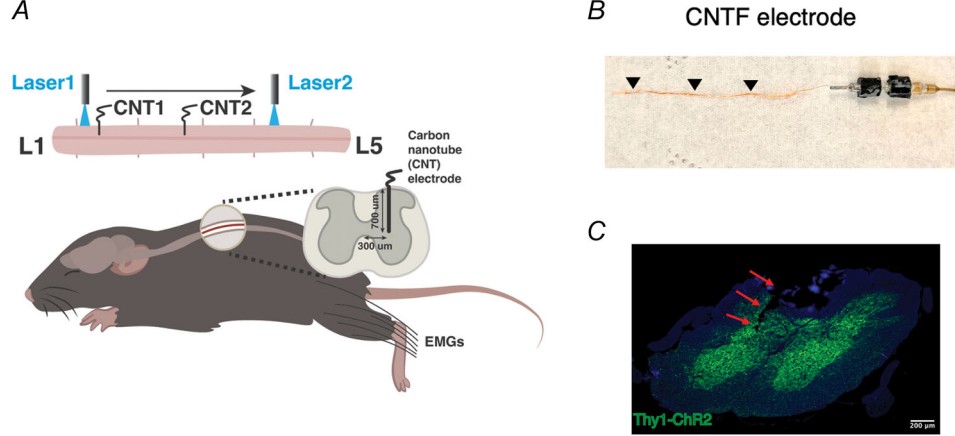

**Figure 1. Experimental set-up.**
*A*, neural and muscle activities were elicited by optogenetic stimulation while the mouse was under anaesthesia. The two carbon nanotube fibre (CNTF) electrodes were typically inserted into L2 and/or L3 spinal segments, ipsilateral to the implanted muscles. *B*, a CNTF electrode, coloured in orange and whose length is marked by triangles, along with its connector (16-channel Samtec attached to ZIF-Clip) on the right. *C*, a histological slice of the spinal cord showing the electrode's route of penetration (marked by red arrows). The CNTF electrode was inserted into the spinal cord from the dorsal surface of the cord. Neurons with Thy1-ChR2 expression were labelled green. [Colour figure can be viewed at wileyonlinelibrary.com]

cord, ipsilateral to the side with implanted muscles, and with the laser fibre tip just touching the spinal surface (Fig. 1A). We have previously verified that this touching would not induce any mechanical stimulation of the spinal cord (Salmas & Cheung, 2023). This laser was moved in a rostral-to-caudal direction, in 200 μm steps, to exhaustively activate all accessible excitatory neuronal networks, so that data from a total of 7 to 18 stimulation loci (mean ± SD: 13 ± 4 loci) were obtained from each animal. For each spinal stimulation locus, we first established the minimum laser power just sufficient to elicit an observable muscle contraction and obtained neural and muscle activities under this condition of single-fibre stimulation at threshold power

(SST condition) (Fig. 2A). Then we increased the stimulation intensity to 120%–150% of threshold power to obtain more activities under the condition of single-fibre stimulation at above-threshold power (SSAT condition). For each spinal location, 5 SST and then 5 SSAT trials were recorded with an intertrial duration of 1000 ms. In each trial, optical stimulation was delivered with square pulses of 5 ms width at 100 Hz over a 200 ms duration.

In the second part of the experiment, a second laser fibre was fixed at the most rostral position of the exposed spinal cord, whereas the first laser fibre would now be at the caudal-most position after the previous rostral-to-caudal spinal mapping. Then we moved the first laser again across stimulation trials, now in the

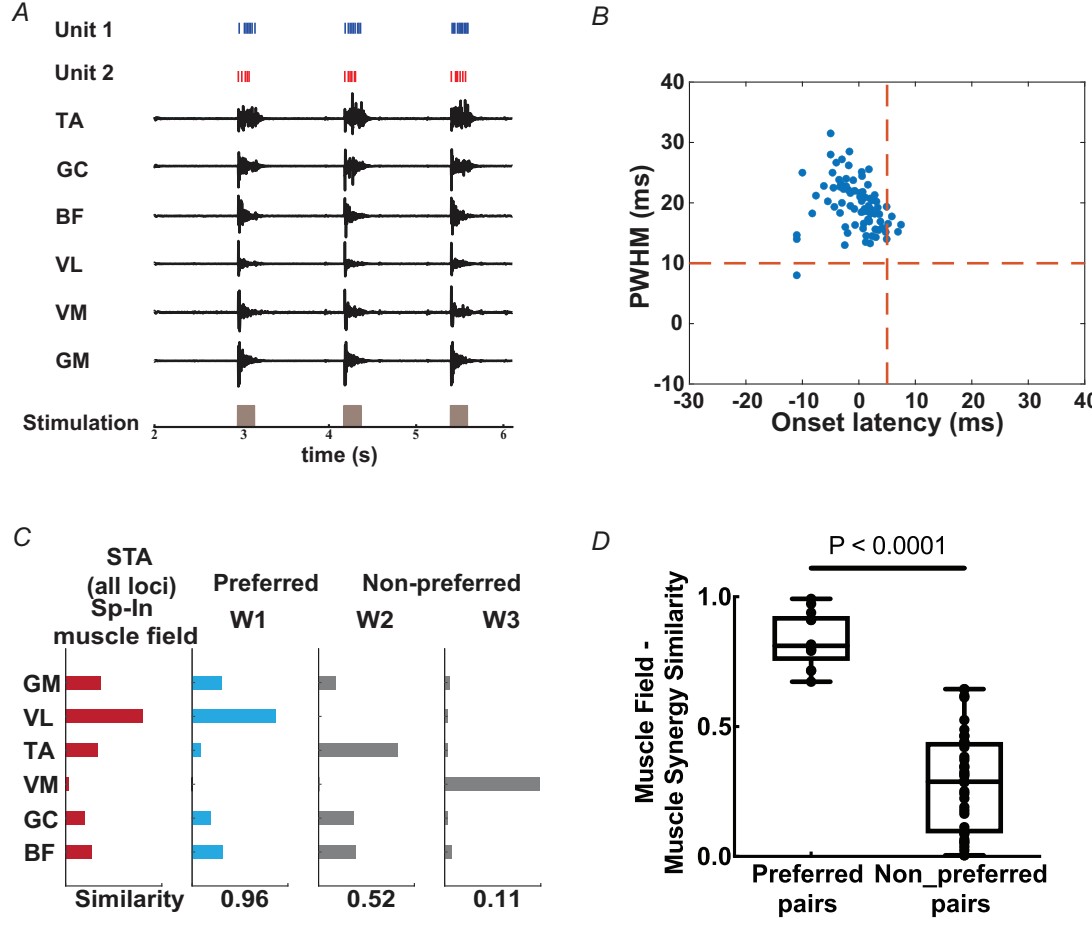

**Figure 2. Muscle fields of interneuronal units could be matched to EMG-derived muscle synergies.**
*A*, raw spikes of two units and raw EMG activities of six hindlimb muscles, temporally aligned with the optogenetic stimulation that elicited them. Data from three trials of the same spinal stimulation locus are shown. *B*, two-dimensional distribution of peak width at half maximum (PWHM) (mean ± SD: 19.6 ± 4.1 ms) and onset latency (mean ± SD: −0.4 ± 4 ms) of post-spike facilitations from all recorded spinal interneurons (13 units from 9 mice). *C*, the muscle field (in red) retrieved by spike-triggered averages (SpTA) could be well matched to one of the EMG-derived preferred muscle synergies W1 (blue). The other two synergies W2 and W3 (grey) could not be well matched to this specific muscle field. The pairwise scalar product value between the muscle field and each synergy is shown at the bottom of the synergy. *D*, the pairwise similarity between the muscle fields and muscle synergies for the preferred pairs (N = 13) is significantly higher than those for the non-preferred pairs (N = 38) (unpaired *t* test, P < 0.0001). The error bars represent the minimum and maximum values of the data. [Colour figure can be viewed at wileyonlinelibrary.com]

caudal-to-rostral direction, in 200 μm steps, whereas the second laser remained still at the rostral-most position. In each trial, we delivered simultaneous stimulations to the two lasers, such that we could obtain neural and muscle activities resulting from pairs of different stimulation sites (Mussa-Ivaldi et al., 1994; Tresch et al., 1999). Before each co-stimulation trial, the threshold power needed of the moving laser was redetermined at its stimulation site, so that activities were obtained under the condition of co-stimulation at threshold power (CoST condition). From each mouse data from 10 to 19 pairs of co-stimulated spinal loci (mean ± SD: 13 ± 3 loci) were obtained.

**Histology.** After the experiment the mouse was perfused transcardially with phosphate-buffered saline (PBS) followed by 4% paraformaldehyde (PFA). The spinal cord was extracted and kept in PFA (4%) at 4°C overnight and then left in sucrose (30%). Slices of the spinal cord were collected in a cryostat at thickness of 10 μm. Slices were then incubated and mounted with 4′,6-diamidino-2-phenylindole (DAPI; Thermo Scientific) on microscope slides and imaged using a confocal microscope (Leica TCS SP8 WLL Inverted Confocal Microscope with Tokai-hit Incubation System). The depth of penetration was determined in FIJI using the Straight Line tool to draw a line along the extent of the penetration. Then the measurement was obtained using the Measure function (Schindelin et al., 2012).

**Spinal electrophysiology recording using CNTF electrodes.** We recorded evoked interneuronal activities from the lumbosacral intermediate grey region using CNTF electrodes. One (for 4 animals) or two (for 5 animals) CNTF electrodes were inserted into two arbitrary locations of the exposed spinal cord, but always at 300 μm lateral to midline, and 700 μm in depth (Fig. 1C). We focused on depths corresponding to Rexed laminae V-VI, as these regions are thought to contain interneurons that contribute to the generation of muscle synergies (Levine et al., 2014). The two CNTFs were typically inserted into the L2 and/or L3 spinal segments. Because each CNTF electrode could detect the activity of one unit (as confirmed by spike sorting; see below), activities of 1–2 units were collected per animal.

## Statistics

We used the unpaired *t* test to reveal any statistically significant differences between the means of two groups of data. The magnitude of significance was measured by the effect size and calculated as Cohen's d (Cohen, 2013). Additionally we employed the t-statistic to evaluate the linear regression model by quantifying the difference between the observed values and their predicted values, assessing whether the predicted values significantly contribute to the model.

## Spike-triggered average of EMG and muscle field

Synchronized neural and muscle activities were recorded at 24.414 kHz using a 96-channel high-throughput acquisition system (PZ5, RZ2 and RS4 of Tucker-Davis Technologies, Alachua, FL, USA), while the laser beam was controlled via a voltage pulse trigger sent from the RZ2 port. We concatenated the raw neural and EMG data across all trials from all stimulated loci and then down-sampled the data to 1 kHz. We performed spike sorting on the down-sampled neural activities using MountainSort4, a Python-based open-source spike sorting algorithm (Buccino et al., 2020; Chung et al., 2017). Muscle activities were initially acquired using primary bandpass filters set to 10–100 Hz. Following acquisition the signals underwent additional pre-processing steps: high-pass filtering with a 50th-order finite impulse response (FIR) filter (cut-off frequency of 50Hz), rectification and low-pass filtering with a 50th-order FIR filter (cut-off frequency of 20 Hz). Motion artifacts in the muscle activities were visually identified and replaced with the 99.9th percentile value of the muscle activity from the same channel. The artefact-removed EMGs of each muscle were then variance normalized (Guo et al., 2022).

SpTA of EMG quantifies how muscle activities respond to neural spike activities and reveals the functional connectivity between the recorded neuron and activity patterns of the motoneuronal pools linked to it (Fetz & Cheney, 1980). For each neuron we isolated and selected spikes that served as reliable triggers for each muscle (Schieber, 2002). All preprocessed EMG activities were segmented from 30 ms before to 50 ms after each spike and averaged across all spikes (Davidson et al., 2007). All units exhibited significant post-spike effects (Poliakov & Schieber, 1998).

For each muscle, we calculated the mean percentage increment of post-spike EMG facilitation by following this procedure: we identified the SpTA peak in the test window 3–16 ms after the trigger, then averaged the values backward and forward until they fell within 2 SDs of the mean during the baseline sample period (30–10 ms before the trigger time) (Davidson et al., 2007). The increment value across the recorded muscles is a vector that is referred to as the neuron's 'muscle field' (Takei & Seki, 2010).

## Classification of spinal interneurons

It has been shown that units in the motor cortex and spinal cord exhibit a broad spectrum of SpTA of muscle activities (Fetz & Cheney, 1980; Fetz et al., 2002). Pre-

vious studies have shown that the peak or trough of SpTA typically appears in a 6-to-16 ms time window. However when other participant units, which indirectly contribute to the muscle activities, emerge before, during or after the spike trigger, the SpTA may exhibit a broader peak width and an earlier onset. This phenomenon is referred to as the synchrony effects of SpTA (Perlmutter et al., 1998). Although the units whose SpTAs exhibit synchrony effects are not considered to have direct control over the corresponding motoneuronal pools, these effects nonetheless represent a substantial indirect contribution to the control of muscles (Perlmutter et al., 1998). Thus we used features of both onset latency and peak width at half maximum (PWHM) as a two-dimensional measurement to quantitatively categorize SpTA effects into pure and synchrony effects. Significant SpTA effects with onset latency greater than 3.5 ms and PWHM less than 7 ms were considered as pure post-spike effects (Schieber & Rivlis, 2005; Takei & Seki, 2010). Units with pure post-spike effects are likely premotor interneurons that synapse directly with the motoneurons. In our data of 81 SpTAs, we recorded smaller-onset latency (mean ± SD: −0.4 ± 4 ms) and lager PWHM (mean ± SD:19.6 ± 4.1 ms), which reflect contributions of synchrony effects from other recruited premotor interneurons or other spinal interneurons (Fig. 2B). Such SpTA results are consistent with properties of the neurons in the area where the electrode was positioned (mediolateral: 300 μm from midline, dorsoventral: mean depth of 561 ± 70 (SD) μm). As a result all recorded units were classified as upstream spinal interneurons with at least two synapses away from the motoneurons.

## Muscle synergy extraction

The non-negative matrix factorization (NNMF) algorithm was used to extract muscle synergies and their activation coefficients from the normalized EMGs (Lee & Seung, 1999). The component weights within each muscle synergy reflect how the activated muscles are co-ordinated. The activation coefficients reflect how a particular synergy was activated over time. If $D$ is the non-negative $m * n$ data matrix comprising n samples of an $m$-dimensional vector, $D$ is modeled as the linear combination of two matrices $W$ and $C$:

$$D = WC^T + R = \sum w_i c_i^T + R \tag{1}$$

where each vector $w_i$ is the $i$th column of $W$, denoting the $i$th muscle synergy, vector $c_i$ is the $i$th column of $C$, the temporal coefficient of $w_i$, and $R$ is the residual that cannot be explained by the model. To determine the number of synergies needed to reconstruct the EMGs, we increased the number of extracted synergies from one to the number of recorded muscles and selected the minimum number of

synergies required for an EMG reconstruction $R^2$ of 80%. The $R^2$ was calculated as follows:

$$R^2 = 1 - \frac{SSE}{SST} \tag{2}$$

$$SST = \sum_{i,j} \left(D_{ij} - mD_i\right)^2; \quad SSE = \sum_{i,j} \left(D_{ij} - \left[WC^T\right]_{ij}\right)^2 \tag{3}$$

where SST is the total sum of squares, $mD_i$ is the average EMG value of the $i$th muscle and SSE is the sum squared error.

The NNMF algorithm was run 100 times for each number of synergies to avoid convergence to a local minimum. Each NNMF run was initialized with random values, for both the muscle-weight and coefficient matrices, obtained from a uniform distribution between 0 and the maximum EMG amplitude.

## Clustering muscle synergies

One major goal of our analysis was to assess the similarity between the muscle fields of the neuronal units and the muscle synergies extracted from the EMGs in each animal and each stimulation condition. To see whether the muscle fields may be preferentially matched to particular types of muscle synergies across animals, we performed a conventional clustering analysis (Yakovenko et al., 2011) to classify all muscle synergies from all mice into groups. *K*-means clustering was performed for all synergies from the SST condition of the nine mice, using the MATLAB function kmeans. The number of clusters was determined by iterating the clustering algorithm with the number of clusters varying from 2 to 10. At each number of clusters the within- and between-cluster Euclidean distances between pairs of synergy vectors were calculated, and the clustering performance was evaluated by the averaged silhouette value (Rousseeuw, 1987). The number of clusters that yielded the first maximum silhouette value was considered as the optimal number of clusters. Within each cluster of synergies, those that could be matched to a muscle field were then identified.

## Similarity between muscle fields and muscle synergies across conditions

For each animal we quantified the pairwise similarity between the muscle fields (or muscle synergies) derived from the different stimulation conditions (SST, SSAT or CoST) using the scalar product between L2-normalized vectors. The muscle fields (or synergies) from the different stimulation conditions were identified separately. Within each condition the muscle fields (or synergies) were matched to those from another condition by maximizing the total scalar product value. The pairwise scalar

products were then averaged across all pairs for each animal to reflect the overall similarity between different sets of muscle fields (or synergies).

### Across-condition modifications of the units' preferred muscle synergies

After we evaluated the similarities for both the muscle fields and muscle synergies across conditions, we observed that for many units their muscle fields under different stimulation conditions differed even though they could still be matched well to muscle synergies of their respective conditions. Thus different synergies were preferentially matched to the muscle fields of the same unit in different conditions. To characterize how these muscle synergies matched to the same unit were modified from one condition to another, we note that previous studies have identified three forms of muscle synergy changes: merging (Cheung et al., 2012; Clark et al., 2010), fractionation (Cheung et al., 2012) and complete alterations (Berger & d'Avella, 2023; Berger et al., 2013). In particular, in human runners specific synergy 'merging patterns' correlated with the energetic efficiency of running, suggesting that any changes in the control strategy implemented by the nervous system could be reflected by muscle synergy changes (Cheung et al., 2020). These prior results prompted us to systematically evaluate the relationships between these muscle synergies that were matched to the muscle fields of the same unit under different stimulation conditions.

For each muscle field in each condition, we first identified the muscle synergy of that condition that matched best with the field (i.e. the field's 'preferred synergy'). We then assessed whether the preferred synergy of the SSAT or CoST condition could be explained by merging multiple synergies of the SST condition that included the same unit's preferred synergy through linear combination. The coefficients of this linear combination were identified by non-negative least squares procedure (implemented by the MATLAB function 'lsqnonneg'), and a synergy was considered to be involved in merging only if its associated combination coefficient was greater than 0.2 (Cheung et al., 2020). Those preferred synergies that could not be explained by merging were characterized as 'unmatched' if the across-condition pairwise scalar product of the preferred synergies was <0.8, or 'preserved' if otherwise.

### Correlation between neural firing rate and activation coefficient

Because previous studies argued that muscle synergies could be represented by spinal interneurons (Takei & Seki, 2010; Takei et al., 2017), we speculated that the firing rate of individual interneurons may correlate with the activation coefficients of muscle synergies. To derive a neuron's firing rate at a spinal stimulation locus we calculated the across-trial average of time-averaged firing rate. For each muscle synergy, we likewise calculated the amplitude of trial- and time-averaged activation coefficient for each stimulation locus. We then calculated the absolute Pearson correlation coefficient between these two variables that varied across the stimulation loci.

For the firing rate of each spinal interneuron, we demonstrated its potential correlation with a synergy's activation coefficient with two different approaches. In the first approach we examined its absolute correlation coefficient with the activation coefficient of the unit's preferred synergy ($C_{corres}$). In the second approach we identified the muscle synergy with the activation coefficient that exhibited the highest absolute correlation coefficient with the firing rate of that interneuron ($C_{prefer}$).

## Results

### Muscle synergies could be matched to muscle fields

We investigated whether the spinal interneurons recruited by spinal optogenetic stimulations underlie the hindlimb muscle co-ordination patterns by comparing the muscle fields with the muscle synergies (Takei et al., 2017). In 13 units recorded from nine mice, stimulations elicited significant SpTA synchrony effects in the recorded muscles (Fig. 2B). For each unit, the level of activation of each muscle was quantified by the muscle's mean percentage increment (Davidson et al., 2007), and the increment values across muscles together comprise the unit's muscle field (Fig. 2C, red bars).

For this first comparison we evaluated the similarity between muscle fields and muscle synergies that were elicited by a single laser beam at the lowest optical power, with the laser positioned at a different spinal locus in each trial (Caggiano et al., 2016). This experimental design maximizes the motor variability acquired from the spinal stimulations while minimizing the volume of neural tissue stimulated at each locus. Similarity was quantified as the scalar product of two paired vectors whose magnitudes were normalized. For each mouse, three or four synergies were extracted from the EMGs, and each muscle field was paired with every muscle synergy. The pair with the highest scalar product was then identified and referred to as the 'preferred pair', and the matched synergy was referred to as the 'preferred synergy' of that particular muscle field (Fig. 2C). Across all 13 units, the muscle fields showed a very significantly higher similarity to the preferred synergies ($N = 13$) than to the non-preferred synergies ($N = 38$) (unpaired *t* test, $P < 0.0001$, Cohen's d = 3.1747; Fig. 2D), with the average similarity to the former at $0.83 \pm 0.1$ (mean ± SD).

To study how the different types of muscle fields or muscle synergies may represent the basic muscle co-ordination patterns encoded in the spinal cord, we first clustered all muscle synergies extracted from the EMGs of the nine mice. All but one cluster of synergies (cluster 8) contained synergies identified from multiple animals (Fig. 3, grey bars). Also all but one synergy cluster (cluster 8 again) included at least one preferred synergy (blue bars) that was matched to the muscle field of a unit from the same animal (red bars). Thus optical stimulations over the lumbosacral spinal cord produced nine muscle synergy types, eight of which could be linked to the muscle fields of spinal units in at least some animals. The unmatched muscle synergies may correspond to muscle fields of other neurons that were not recorded in the experiment.

## Muscle fields varied across stimulation conditions

If the muscle field here represents the connectivity between the recorded spinal interneuron, likely an upstream spinal interneuron (Fig. 2B) (see Discussion),

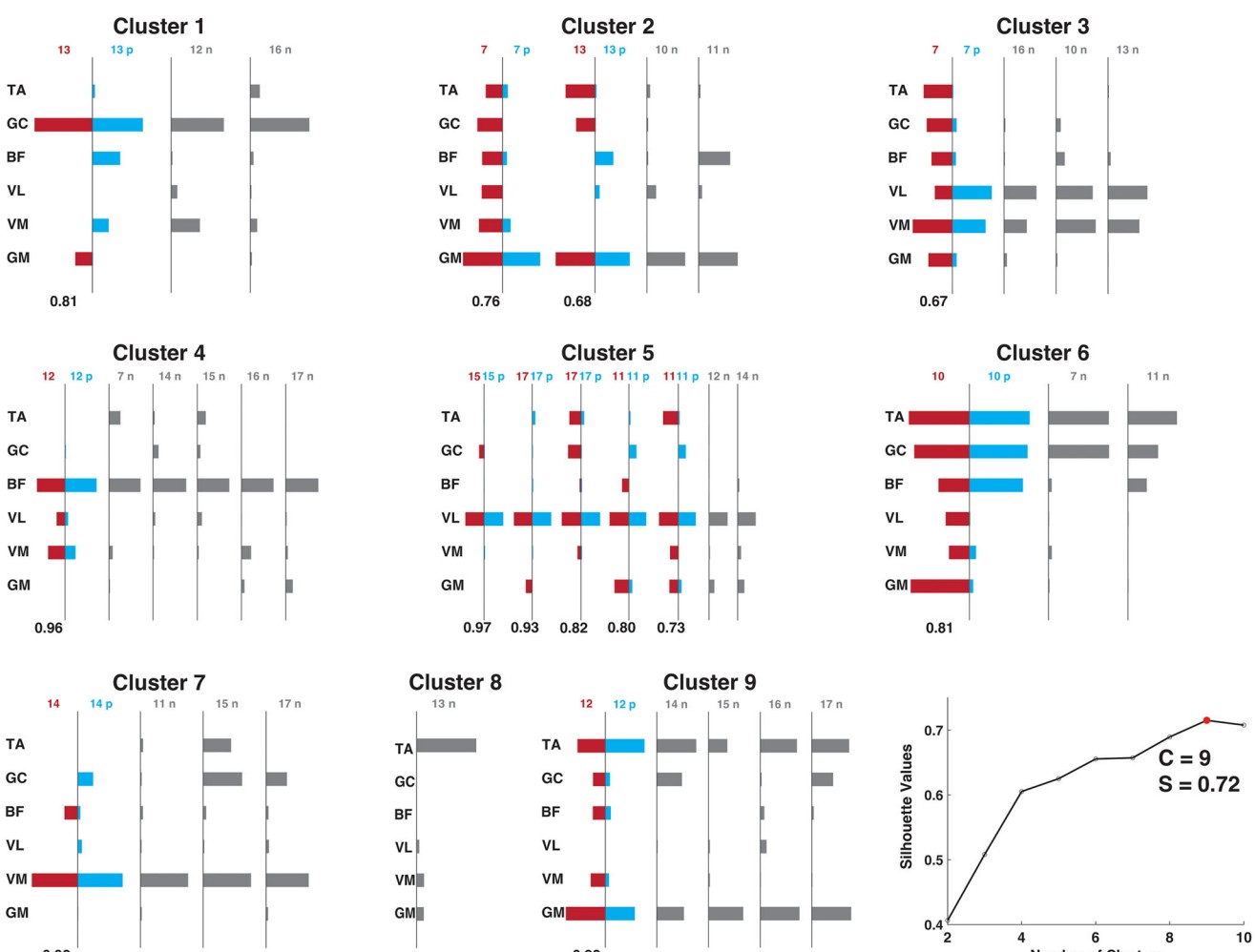

**Figure 3. Muscle synergies from different clusters could be matched to muscle fields of different upstream interneurons.**
This figure displays nine clusters of muscle synergies, with the horizontal bars representing either muscle synergies or the muscle fields matched to the synergies. Muscle synergies that are best matched to a muscle field are shown in blue, while their corresponding muscle fields are shown in red. Muscle synergies not matched to any muscle field are depicted in grey. On top of each muscle synergy or muscle field vector, the vector's corresponding mouse number and a preference indicator are provided: indicator 'p' indicates that the synergy was preferentially paired with the muscle field, while indicator 'n' indicates that the synergy was not paired with any recorded muscle field. The number of synergy clusters ($n = 9$) was identified based on the number that yielded the highest silhouette value, as shown in the plot at the bottom right (C = 9, S = 0.72). [Colour figure can be viewed at wileyonlinelibrary.com]

and the motoneuronal pools of the muscles, we wondered whether this connectivity would remain consistent in different stimulation conditions. We therefore proceeded to stimulate the spinal cord in three different conditions: SST (single laser, threshold power), SSAT (single laser, above-threshold power) and CoST (double-laser co-stimulations, threshold power). The across-condition consistency of the muscle fields was evaluated by calculating the pairwise scalar product between the SST muscle field and the SSAT or CoST muscle fields of the same unit. Figure 4A and B show examples of SSAT and CoST muscle fields from two different mice, respectively. In Fig. 4A the SST muscle field of unit 1 (red bars, upper panel), with strong activations in BF, VL and VM, was sparser than its corresponding SSAT field (red bars, lower panel), which consisted of activations of all six muscles. In Fig. 4B the SST muscle field of unit 2 contained stronger VL activation, but this component was suppressed in CoST; also TA was activated in SST but not in CoST. In all the 14 across-condition pairs of muscle fields (7 from SST *vs.* SSAT, 7 from SST *vs.* CoST), only 29% (4 pairs) exhibited across-condition consistency (scalar product $\geq 0.78$). Interestingly in contrast to the muscle fields, among all the across-condition muscle synergy pairs, 90% of the pairs exhibited consistency.

The above changes in the muscle fields across conditions led us to investigate whether the muscle synergies could be equally well matched to the muscle fields in different conditions. We evaluated the scalar product for all the field-synergy pairs in each condition and divided the pairs into best-matched and unmatched pair groups as before. Scalar product values of the best-matched pairs in SST (Fig. 4C, blue dots) were not significantly different from those of the best-matched pairs in SSAT (red dots). Similarly the best-matched pairs in SST (Fig. 4D, blue) and CoST (red dots) showed no significant difference in scalar product. However the within-condition similarity of the best-matched field-synergy pairs in all conditions was significantly higher than the across-condition similarity of muscle fields. Specifically the similarity of the best-matched field-synergy pairs for both SST and SSAT was significantly higher than the similarity between the SST and SSAT fields (unpaired $t$ test: in SST: $P = 0.0188$, $dF = 12$, tstat $= 2.7157$; in SSAT: $P = 0.0570$, $dF = 11$, tstat $= 2.1259$). Analogous similarity results were obtained also in the SST-*vs.*-CoST comparison (unpaired $t$ test: in SST: $P = 0.0046$, $dF = 12$, tstat $= 3.4793$; in CoST: $P = 0.0174$, $dF = 12$, tstat $= 2.7556$). This result suggests that different stimulation conditions did not affect the overall matching quality between the muscle fields and muscle synergies despite the across-condition changes of the muscle fields of the same units.

## Changes in muscle fields reflected by three forms of synergy modifications

Because the muscle fields changed across stimulation conditions (Fig. 4), we wondered whether the changes occurred through specific patterns so that even after the change from SST to SSAT or CoST, the unit's muscle field could still be matched to one of the muscle synergies of the new condition reasonably well.

To uncover these patterns of muscle field changes, we examined the preferred synergies of the same units from the different conditions. We noticed three forms of synergy modification that may account for how the preferred synergy of a unit was altered from SST to SSAT or CoST. In the first form, called 'Merging', the SST-preferred synergy was linearly combined with another SST synergy of the same mouse to produce the SSAT or CoST preferred synergy. In the example shown in Fig. 5A unit 1's muscle field in SST was matched to its preferred synergy, $S1_{SST}$, with a similarity of 0.88. However in SSAT $S3_{SSAT}$ emerged as the unit's preferred synergy, with a similarity of 0.66 between itself and the unit's SSAT muscle field. Notably $S3_{SSAT}$ could be reproduced by combining the $S1_{SST}$ and $S3_{SST}$, suggesting that the preferred synergy in SSAT was a composite of the preferred and an unmatched synergy in SST. The form of synergy modification through merging is analogous to the phenomenon of merged muscle synergies described in earlier studies (Cheung et al., 2012, 2020; Clark et al., 2010).

In the second form, called 'Unmatched', the SST and SSAT/CoST preferred synergies were totally different to the extent that merging could not account for the difference. In the example in Fig. 5B, in SST condition $S4_{SST}$ was the preferred synergy for unit 1, with a similarity of 0.91. In SSAT $S3_{SSAT}$ was the preferred synergy, but $S3_{SSAT}$ did not correspond to $S4_{SST}$ and could not be reproduced by merging $S4_{SST}$ with any other synergy from SST.

In the third form, called 'Preserved', the preferred synergies in SST and SSAT/CoST were nearly identical. In the example in Fig. 5C the two preferred synergies from SST and SSAT had a similarity of 0.98.

We proceeded to quantify the prevalence of these three forms in the SST-to-SSAT and SST-to-CoST transitions, respectively. In the SST-to-SSAT transition the merging form (form 1) dominated, accounting for nearly 80% of the changes in SSAT (Fig. 5C, black). However in the SST-to-CoST transition the unmatched form (form 2) dominated, explaining almost all of the changes (Fig. 5C, blue). Note that in the latter transition no preserved form (form 3) was identified. Overall these three forms of synergy modifications highlight how the muscle field of a unit may change while still maintaining correspondence with a preferred muscle synergy as the stimulation condition changes.

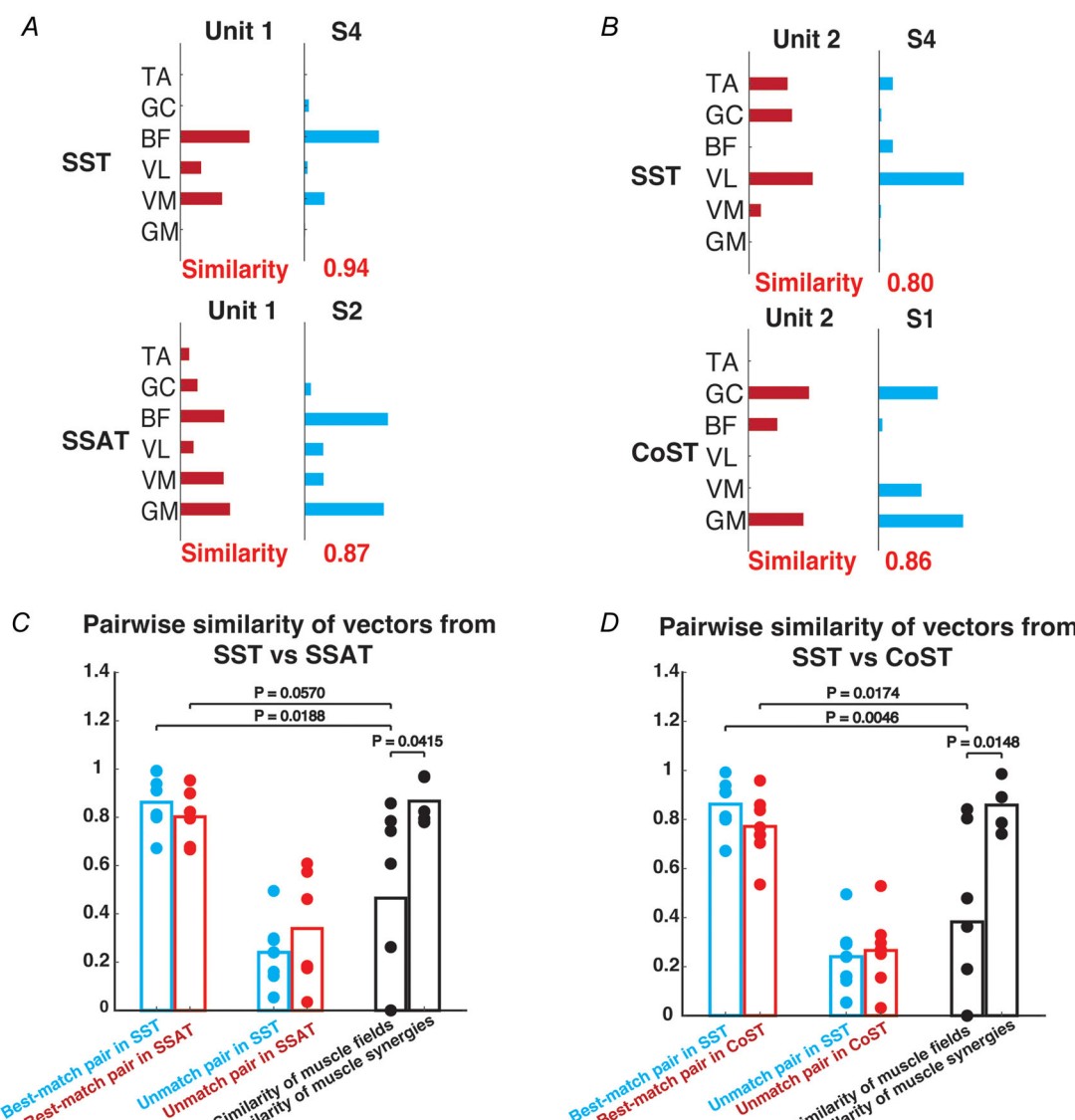

**Figure 4. Pairwise similarity between muscle fields and muscle synergies, between muscle synergies of different conditions and between muscle fields of different conditions (single-fibre stimulation at threshold power (SST), single-fibre stimulation at above-threshold power (SSAT) and co-stimulation at threshold power (CoST)).**

*A*, an example showing how the muscle field of a unit (unit 1, red) changed from SST to SSAT even though a muscle synergy from each condition (synergy S4 in SST, S2 in SSAT; blue) could still be well matched to the unit's muscle field in that same condition. The similarity value is indicated at the bottom of the synergy bar. *B*, analogous to (*A*), muscle field of unit 2 (red) changed from SST to CoST, but a synergy could still be well matched to the muscle field of both conditions despite the change. *C*, pairwise similarity between muscle synergies of SST and SSAT is significantly higher than that between muscle fields of SST and SSAT (unpaired *t* test, *P* = 0.0415, d*F* = 10, tstat = 2.3372), as shown by the black bars. Similarities between muscle fields and synergies in SST (or SSAT) are shown in blue (or red). 'Best-match pairs' refer to the pairings of muscle fields with their corresponding preferred synergies, and 'unmatch pairs' refer to the pairings of muscle fields with the rest of the synergies (non-preferred synergies). Both the within-condition similarity of the best-matched field-synergy pairs in SST and that in SSAT are significantly higher than the across-condition similarity between the muscle fields of the two conditions (unpaired *t* test: *P* = 0.0188, d*F* = 12, tstat = 2.7157; *P* = 0.0570, d*F* = 11, tstat = 2.1259). *D*, analogous to (*C*), but showing data for the SST *vs*. CoST comparison (unpaired *t* test: *P* = 0.0046, d*F* = 12, tstat = 3.4793; *P* = 0.0174, d*F* = 12, tstat = 2.7556). [Colour figure can be viewed at wileyonlinelibrary.com]

## Lack of correlation between unit firing rate and the coefficients of the unit's preferred synergy

In the muscle synergy model, each time-invariant synergy is recruited by a time-varying activation coefficient. If a muscle synergy can be matched to the muscle field of a neuronal unit, it is natural to ask whether the activation coefficients of the synergy correlate with the firing rate of the synergy's corresponding neuronal unit. We began this inquiry by examining how the synergies' coefficients varied across the spinal stimulation loci. As an example in one mouse (no. 19_12) four muscle synergies could explain the EMGs elicited from all loci (Fig. 6A), but the coefficients of these synergies exhibited different activation amplitudes from the rostral to caudal

loci (Fig. 6B, blue curves). For instance coefficient C1 (for synergy S1) was highly activated when the caudal lumbosacral spinal cord was stimulated, whereas C3 was highly activated when the rostral portion was stimulated.

To test whether the firing rate of a unit correlates with the activation coefficient of its preferred synergy, for each locus we calculated the average value of unit's trial-averaged firing rate (Fig. 6B, red curve) and the trial-averaged amplitudes of all activation coefficients (blue curves). In the experiment shown in Fig. 6A and B synergy S4 was the preferred synergy of the unit (whose firing rate is shown in the top panel of Fig. 6B). However activation coefficient C4 showed a very weak correlation with the unit's firing rate (r = −0.47); instead the unit

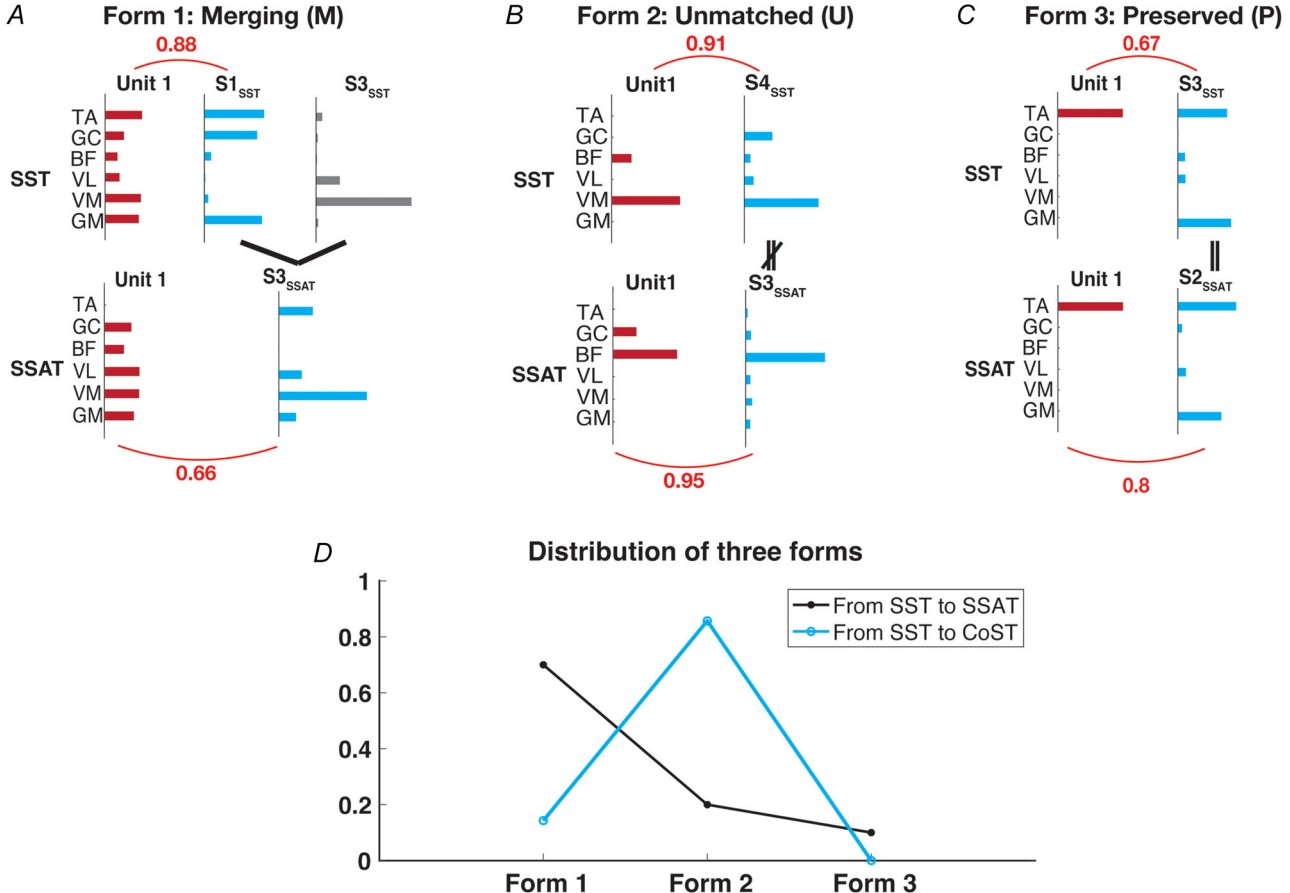

**Figure 5. Changes in muscle fields explained by three forms of modifications of the preferred muscle synergies.**
*A*, muscle fields of unit 1 in single-fibre stimulation at threshold power (SST) and single-fibre stimulation at above-threshold power (SSAT) could be well matched to the preferred synergy S1$_{SST}$ and S3$_{SSAT}$, respectively. Form 1: merging ('M') – The altered muscle field of unit 1 in SSAT can be explained by merging preferred synergies S1$_{SST}$ and an additional synergy S3$_{SST}$, resulting in preferred S3$_{SSAT}$. *B*, form 2: unmatched ('U') – muscle fields of unit 1 were matched to two distinct synergies, S4$_{SST}$ and S3$_{SSAT}$ in SST and SSAT, respectively. *C*, form 3: preserved ('P') – muscle fields of unit 1 were matched to two similar synergies, S3$_{SST}$ in SST and S2$_{SSAT}$ in SSAT. *D*, the prevalences of the three forms of modification differed between transitions from SST to SSAT and from SST to CoST. Form 1 (merging) dominated in the SST-to-SSAT transition. Form 2 (unmatched) dominated in the SST-to-CoST transition. [Colour figure can be viewed at wileyonlinelibrary.com]

correlated best with C2 (r = 0.52). Thus a unit's firing rate may not correspond with the coefficient of its preferred synergy.

We then proceeded to plot the trial-averaged coefficient amplitudes of the units' preferred synergies ($C_{corres}$) against the trial-averaged firing rate of the units, for all units recorded at SST over all stimulation loci, and found no significant correlation between these two variables (t-statistic: $P = 0.123$, Rsq = 0.025, d$F$ = 96; Fig. 6C). When the amplitudes of the coefficient that showed the strongest correlation with unit firing rate ($C_{prefer}$) were plotted for all units instead, there was a statistically significant but weak positive correlation between the coefficient and unit firing rate (t-statistic: $P = 0.00633$, Rsq = 0.075, d$F$ = 96; Fig. 6D).

## Discussion

### Neural representations of hindlimb muscle synergies

In this study we demonstrated that the muscle fields of spinal interneurons corresponded well to the hindlimb muscle synergies derived from multimuscle EMG recordings (Figs. 2 and 3). Our findings from the anaesthetized mice are consistent with previous frog (Hart & Giszter, 2010) and monkey (Takei et al., 2017) studies that demonstrated, using concurrent spinal neuronal and EMG recordings, the roles of spinal interneurons in encoding behavioural muscle synergies. Specifically in Takei et al. (2017) synergies for the monkey forelimb behaviours were attributed to the last-order premotor interneurons (PreM-INs). Here given that the units we obtained are likely interneurons upstream of the

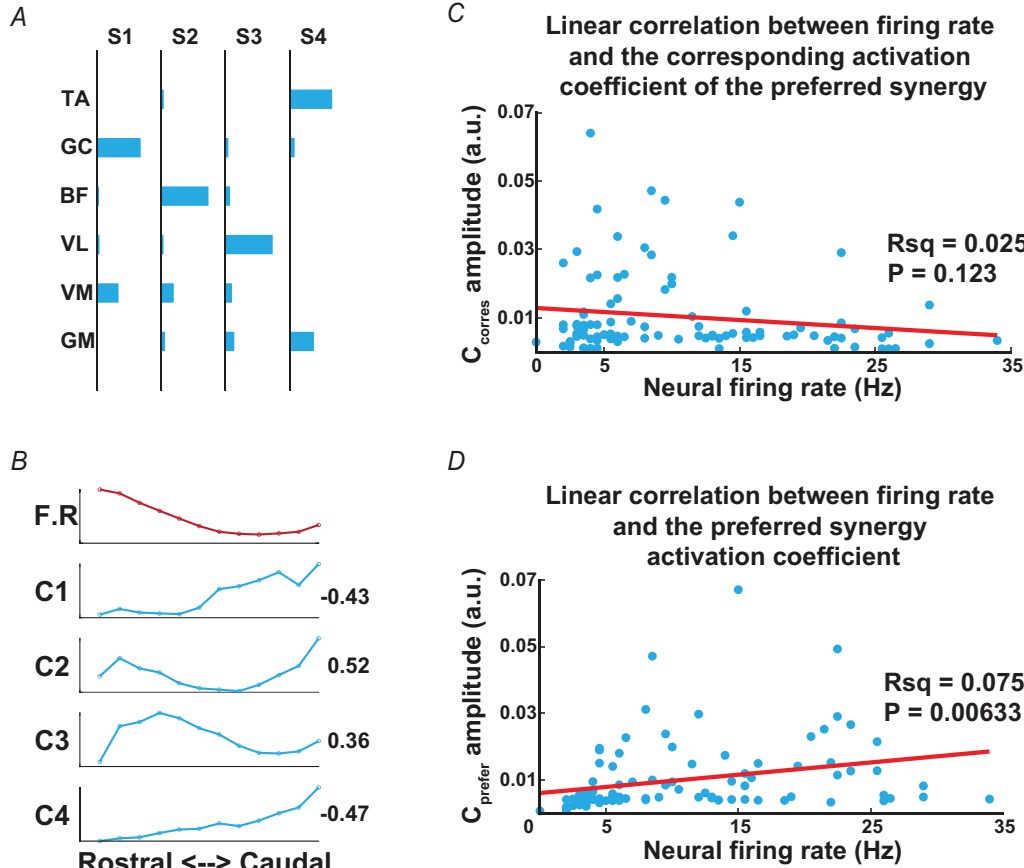

**Figure 6. Correlation between firing rates and synergy activities.**
*A*, four synergies were extracted from EMGs concatenated from the recordings of all stimulation loci of 'mouse_19_12'. *B*, the trial-averaged neural firing rate of a unit from the same mouse in (*A*) (F. R, shown in red) is plotted across the stimulated spinal loci from rostral to caudal. Below the amplitudes of trial-averaged synergy activities for the four synergies (C1–C4, shown in blue) across the same spinal loci are displayed. The correlation coefficient between the neural firing rate and the synergy activity for each pair is indicated to the right of each axis. *C*, linear correlation between the averaged neural firing rate and the amplitude of corresponding activation coefficient ($C_{corres}$) of the preferred synergy (t-statistic: $P = 0.123$, Rsq = 0.025, d$F$ = 96). *D*, linear correlation between the averaged firing rate and the amplitude of the activation coefficient with maximum correlation coefficient value ($C_{prefer}$) (t-statistic: $P = 0.00633$, Rsq = 0.075, d$F$ = 96). [Colour figure can be viewed at wileyonlinelibrary.com]

PreM-INs (Fig. 2B), our results extend the previous findings further in suggesting that robust hindlimb muscle synergies are specified by a network that includes the PreM-INs, other upstream spinal interneurons and potentially the sensory neurons (Payne et al., 2020) as well.

It is worth noting that the number of muscle synergy clusters in our study (9 clusters, Fig. 3) is larger than that in Takei et al. (2017) (3 clusters). This difference may be due to our experimental set-up. Our systematic stimulation of the entire length of the lumbosacral cord would produce great variability of the evoked EMGs, thereby leading to a greater number of synergy clusters. Although some muscle synergies could not be matched to the muscle fields (e.g. cluster 8 in Fig. 3), this does not imply the absence of their neural representations. Indeed the diverse connectivity patterns between the upstream spinal interneurons, the PreM-IN network and the motoneuronal pools (Fetz et al., 2002) may form a repertoire of rudimentary muscle synergies (Salmas & Cheung, 2023) from which all behavioural muscle synergies are derived.

Although our study focuses on the functional encoding of muscle synergies in the spinal interneurons, the question of whether the neurons that encode the different synergies exhibit a topographical organization relative to that of the motoneuronal pools of the different muscles remains intriguing. Anatomical studies have established that the motoneuronal pools of the hindlimb muscles are organized as overlapping anteroposterior columns spanning different numbers of spinal segments (Mohan, Tosolini, & Morris, 2014). A recent work has shown that the premotor interneurons that activate motor synergies of the distal muscles (including the gastrocnemius) have a widespread distribution across multiple spinal segments (L2 to L5) and Rexed laminae, though they are concentrated in the deep medial dorsal horn (Levine et al., 2014). Consistent with this result, stimulation studies in the frog (Saltiel et al., 2005, 2016) and mouse (Caggiano et al., 2016; Salmas & Cheung, 2023) further demonstrated that the spinal loci whose focal stimulation recruited the same muscle synergy could have a very wide distribution that spans multiple spinal segments. The distributions of the different muscle synergies also overlap extensively. Together, these results suggest a complex spatial organization in which interneurons for the different muscle synergies are intermingled and distributed across segments, with unclear relationships with the motoneuronal distributions of the muscles co-ordinated by these synergies.

As our results argue, the encoding of muscle synergies likely depends not only on the connectivity between individual interneurons and the motoneurons, but also on their connectivity patterns with the sensory neurons and other premotor interneurons within the spinal circuits. This network-level complexity implies that the representation of synergies may be altered under different conditions. Likewise the synergy topography that reflects the distributions of the PreM-INs and upstream interneurons capable of recruiting the muscle synergies may also adapt when the spinal cord is subjected to different afferent and descending stimulations. Systematic mapping of interneuron populations combined with connectivity analysis will be critical to resolve whether the spinal topography of the synergy-encoding interneurons follows a more-or-less invariant map, or emerge dynamically from the interactions between neurons in the spinal network and network inputs.

## Upstream spinal interneurons may encode muscle synergies in a state-dependent manner

Our results further showed that the muscle fields of the same upstream spinal interneurons changed across stimulation conditions. Specifically from SST to SSAT the new muscle field could often be explained by merging the synergy matched to the original muscle field with another muscle synergy (Fig. 5). One possibility that may explain this finding is that other premotor interneurons are recruited as the stimulation intensity increases, thus leading to the merging of two muscle fields through a hypothetical mechanism that is illustrated in Fig. 7. In this figure the motoneuronal pools of the recorded muscles (M1 to M4, blue squares) can be directly recruited by both excitatory (green rectangles) and inhibitory PreM-INs (yellow rectangles) and indirectly by upstream spinal interneurons (N1–N6, green and yellow circles). In the SST condition (Fig. 7A) stimulation of the excitatory interneuron N1 at threshold power activates the recorded interneuron N3, which in turn activates the premotor interneuron PreM-IN$_\alpha$, a neuron that co-ordinates activities in M2, M3 and M4. In the SSAT condition, when N1, and therefore N3, is stimulated at a high intensity (Fig. 7B), another excitatory premotor interneuron nearby, PreM-IN$_\beta$, is recruited through an axonal branch of N3. The original muscle fields of PreM-IN$_\alpha$ and PreM-IN$_\beta$ are then merged together to become the new muscle field of N3 as both PreM-INs are coactivated by N3. Because inhibitory interneurons may also underpin any potential negative components in muscle synergies (Guo et al., 2024; Takei et al., 2017), under SSAT other inhibitory PreM-INs (PreM-IN$_\gamma$ in Fig. 7B) may also be directly or indirectly activated by the upstream interneurons to result in a more complex muscle field for N3. Note that in this schematic, the added muscle field from PreM-IN$_\beta$ could be independently evoked under SST by the upstream interneuron N4, thus consistent with our observation that the merged muscle synergies for the SSAT muscle field could always be identified from the EMGs of SST.

On the other hand, under the CoST condition when an additional spinal locus was co-stimulated simultaneously, multiple upstream interneurons with interacting downstream connections are engaged, thus resulting in the original recorded interneuron being associated with a different muscle field. In our hypothetical network (Fig. 7C) the inhibitory PreM-IN$_\theta$ is recruited, through activation of N5, by the additional stimulation. This PreM-IN$_\theta$ produces strong inhibition on the motoneurons for M2, so that when N1, N3 and N5 are coactive, EMG can only be produced in M3 and M4. This effectively changes the muscle field of N3 from M2, M3 and M4 to M3 and M4, with the latter muscle

group being also independently accessible (e.g. under SST condition) through N2, N6 or PreM-IN$_\gamma$ in the network. Our example here illustrates how the same upstream interneuron can be associated with another muscle field as the stimulation condition changes from SST to CoST.

The two scenarios above illustrate how the muscle field of the same upstream interneuron can be matched to different muscle synergies or their combinations after the stimulation condition changes. These patterns of muscle field changes likely result from the cooperative interactions between different subnetworks that involve the upstream spinal interneurons, PreM-INs and motoneurons with their interactions being dynamically

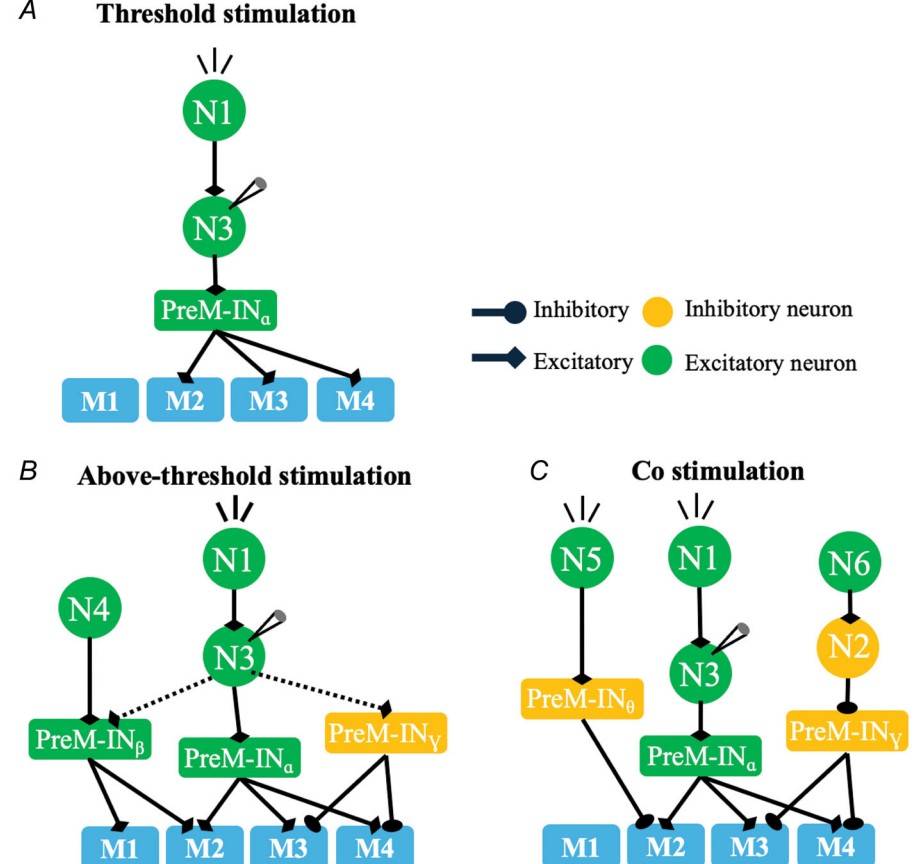

**Figure 7. Hypothetical connectivity patterns between the spinal interneurons and the motoneuronal pools that may account for state-dependent synergy encoding of the upstream spinal interneurons.**
*A*, threshold stimulation (single-fibre stimulation at threshold power (SST)): N1 represents the neuron stimulated directly by the laser beam during the recording trial. N3 is the neuron recorded by the carbon nanotube fibre (CNTF) electrode. The black arrows indicate the connectivity between N3, premotor interneurons (PreM-IN) and the motoneuronal pools of the muscles. Here PreM-IN$_\alpha$, activated by N3, in turn co-ordinates a muscle synergy consisting of muscles M2 to M4. *B*, at above-threshold stimulation (SSAT), the 'merging' synergy modification form (form 1) dominates. Here a neighbouring excitatory premotor interneuron PreM-IN$_\beta$ and an inhibitory premotor interneuron PreM-IN$_\gamma$ are recruited as a result of stronger activations of N1 (and hence, N3). The combined effect of these additional recruitments leads to the merging (i.e. linear combination) of the synergies originally encoded by PreM-IN$_\alpha$, PreM-IN$_\beta$ and PreM-IN$_\gamma$. *C*, in co-stimulation (CoST), modification form 2 ('unmatched') dominates. Here N1 and N5 represent the neurons co-stimulated by the two laser fibres, and PreM-IN$_\theta$ is an inhibitory premotor interneuron that inhibits muscle M2. As a result of such co-stimulation a distinct muscle coactivation pattern (M3 and M4) is associated with the recorded N3. This new pattern can also typically be retrieved by activating N6. [Colour figure can be viewed at wileyonlinelibrary.com]

modulated by supraspinal, propriospinal and sensory inputs to induce time-dependent muscle field changes. Our interpretation suggests that upstream spinal inter-neurons may encode muscle synergies in a manner that is dependent on the activity state of the spinal system.

It is reasonable to wonder to what extent our muscle field results and our interpretation summarized above, all based on data collected from anaesthetized animals, may still hold in naturally behaving animals. Anaesthesia is known to suppress a wide range of neural activity and may affect synaptic transmission. However the neuronal activities measured in our experiments were elicited through optical stimulation, and even at the minimum power of stimulation we were able to retrieve interneuronal muscle fields that matched with the EMG-derived synergies, a result that is consistent with prior recordings obtained during voluntary movement when the animal was not under anaesthesia (Takei et al., 2017). Thus we think the use of anaesthesia should have a minimal impact on the interpretation of our results.

### Stability of muscle synergies encoded by the spinal cord

Even though the muscle fields of the interneurons changed across stimulation conditions, the muscle synergies themselves extracted from the EMGs remained stable in all three conditions overall (Fig. 4) despite occasional instances of synergy merging when the stimulation condition changed (Fig. 5A). In previous studies muscle synergies were identified as robust motor modules that remained stable across locomotor behaviours (Hart & Giszter, 2004). For instance a few basic muscle synergies can account for the variance in human motor patterns during activities such as walking and running (Cappellini et al., 2006) even though the spinal neuronal network surely adopts different states for different behaviours (Pearson & Collins, 1993). Our findings align with the substantial body of evidence suggesting that the CNS utilizes the same set of muscle synergies to generate diverse motor behaviours by modulating their temporal scaling (d'Avella & Bizzi, 2005; Bizzi & Ajemian, 2015; Bizzi et al., 2008; Chvatal & Ting, 2013).

What then are the neuronal underpinnings of selecting the stable muscle synergies? Given that the upstream spinal interneurons have indirect and more dynamic connections to the motoneuronal pools due to inputs from the sensory afferents and other excitatory and inhibitory interneurons (Giszter, 2015), it is plausible that the down-stream PreM-INs, as last-order interneurons, maintain relatively state-independent muscle fields that account for the stable muscle synergies (Levine et al., 2014).

It has been recently proposed that the spinal network encodes a set of default locomotor synergies as 'common core infrastructure' (Desrochers et al., 2019; Sun et al., 2022) that is subject to modifications from descending and sensory inputs, so that the default synergies can be fine-tuned (Cheung et al., 2024; Yang et al., 2019) or even merged (Cheung et al., 2020) for different behaviours. It is tempting to speculate that although the PreM-INs are responsible for structuring the default common core, the upstream spinal interneurons shape and select the final muscle synergies expressed during behaviours, as these interneurons assume different states induced by the changing descending and sensory inputs. Future recordings that explicitly target multiple layers of spinal interneurons, including the PreM-INs, should permit a test of this hypothesis.

### Flexible encoding of neuromotor modules for robust motor behaviours

Previous studies have consistently shown that the neural encoding of various movement parameters is state-dependent, reflecting the adaptability of the neuro-motor system to changes in the environment and task demands. For instance the firing pattern of neurons in the motor cortex and area 5 was dependent on the target direction, hand velocity and position (Ashe & Georgopoulos, 1994), indicating that neural encoding can be shifted by various task commands. Similarly spinal neurons were thought to exhibit state-dependent activities that contributed to the modulation of movement patterns under different sensory inputs (Tresch et al., 1999). A more recent study showed that the firing rate of speed-encoding neurons exhibited distinct firing patterns during forward and backward forelimb cycling, respectively (Saxena et al., 2022). The state-dependent muscle field encoding of the upstream spinal inter-neurons demonstrated here implies that muscle synergies can be flexibly selected, fine-tuned or merged to meet the demands of different motor tasks, or to adapt to changes in the biomechanical requirements of the same task resulting, for instance, from changes in the dynamic environment of the workspace (Shadmehr & Mussa-Ivaldi, 1994). This adaptability ensures that, regardless of the specific behavioural context, the neuro-motor system can generate stable and reliable movements by relying on the same set of upstream spinal interneurons that can activate different synergy sets or merge different synergies depending on the network state. Such flexibility that arises from state-dependent muscle field encoding is crucial for maintaining robust motor behaviours in an ever-changing environment with the most parsimonious set of neuronal network.

## Lack of correlation between neural firing rate and synergy coefficient

In an earlier study (Hart & Giszter, 2010) spinal neurons in the intermediate zone were found to exhibit firing patterns that had a direct relationship with the activation coefficients of muscle synergies, referred to as 'motor primitives' in that study. Although these neurons showed significant SpTA effects on muscle groups that matched well with the muscle synergies identified, it remains unclear in that study whether the activation coefficient of the muscle synergy that was matched to a unit also correlated temporally with the firing rate of the same unit. On the other hand, our results here (Fig. 6C) suggest that individual interneurons whose muscle fields exhibit high spatial similarity to muscle synergies may not fire in a way that correlates with the activation coefficients of their corresponding synergies. This implies that the activation coefficient of a synergy may reflect the firing of not a single neuron, but possibly of a population of neurons, perhaps with similar muscle fields. This possibility was also raised by Takei et al., who demonstrated a rough correspondence ($R^2 = 0.61$) between the coefficient trajectory in the muscle synergy space and the neuronal firing trajectory, but only after the firing rates of PreM-INs with similar muscle fields were averaged (Takei et al., 2017).

The limited number of recorded units here has not allowed us to identify distinct neuronal populations responsible for structuring the synergies and generating the synergies' temporal coefficients, respectively. It has been proposed that the network that shapes the temporal coefficient of a synergy is not only upstream of the PreM-INs, but may also overlap with subnetworks responsible for shaping the coefficients of other synergies, and involve multiple regions within the motor control hierarchy (Cheung & Seki, 2021). These complexities likely contribute to the lack of correlation between unit firing rate and the coefficient of the unit's corresponding synergy observed here. Future studies that aim to relate neural manifolds (Gallego et al., 2020, 2018; Safaie et al., 2023) in spinal neural populations and muscle synergy activities may eventually shed light on the neuronal origin of the synergies' activation coefficients.

## Conclusion

Direct optical stimulations to the spinal cord have revealed neural representations of hindlimb muscle synergies at upstream spinal interneuronal level. Further the synergies' neural representations were found to be state-dependent, as demonstrated by the varied muscle fields associated with the same neurons in different stimulation conditions. State-dependent encoding of muscle synergies may permit the same neuronal networks to select, fine-tune or merge different muscle synergies to produce stable motor behaviours in contexts with different biomechanical demands.

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

## Additional information

### Data availability statement

Preprocessed EMG, neural activities and spike-triggered averages obtained in this study will be made available on request. The data presented in the figures and the MATLAB scripts used to generate those results are available at https://github.com/betty-hbr/Mouse_CNT

### Competing interests

The authors declare no competing interests.

### Author contributions

H.B. and P.S. conceptualized the work, performed experiments, acquired and analysed data, interpreted the results and wrote the manuscript. J.Z. and X.D. assembled CNTF electrodes and tested them in experiments. V.C.K.C. contributed to the conceptualization of the work, interpreted the results and revised the manuscript.

### Funding

The work described in this paper was supported by grants from the Research Grants Council of the Hong Kong Special Administrative Region, China (project numbers: R4022-18, N_CUHK456/21, 14 114 721 and 14 119 022), and from The Chinese University of Hong Kong (Faculty of Medicine Direct Grants numbers: 2020.095 and 2021.065; Research Committee Grant for Impact Cases number C7), to V.C.K.C.

### Keywords

electrophysiology, locomotion, muscle activity, muscle synergies, optogenetic stimulation, spinal cord

## Supporting information

Additional supporting information can be found online in the Supporting Information section at the end of the HTML view of the article. Supporting information files available:

**Peer Review History**

