## [Peer Review History · The Journal of Physiology]

State-Dependent Neural Representations of Muscle Synergies in the Spinal Cord Revealed by Optogenetic Stimulation

Borong HE, Paola SALMAS, Jing ZHANG, Xiaojie DUAN, and Vincent C. K. Cheung
DOI: 10.1113/JP288073

Corresponding author(s): Vincent C. K. Cheung (vckc@cuhk.edu.hk)

Review Timeline:

Submission Date:	07-Nov-2024
Editorial Decision:	09-Dec-2024
Revision Received:	23-May-2025
Editorial Decision:	09-Jun-2025
Revision Received:	23-Jun-2025
Accepted:	03-Jul-2025

Senior Editor: Richard Carson

Reviewing Editor: Matthew Fogarty

Transaction Report:

Dear Dr Cheung,

Re: JP-RP-2024-288073 "State-Dependent Neural Representations of Muscle Synergies in the Spinal Cord Revealed by Optogenetic Stimulation" by Borong HE, Paola SALMAS, Jing ZHANG, Xiaojie DUAN, and Vincent Cheung

Thank you for submitting your manuscript to The Journal of Physiology. It has been assessed by a Reviewing Editor and by 2 expert referees and we are pleased to tell you that it is potentially acceptable for publication following satisfactory major revision.

REVISION CHECKLIST:

Please upload two versions of your manuscript text: one with all relevant changes highlighted and one clean version with no

changes tracked. The manuscript file should include all tables and figure legends, but each figure/graph should be uploaded as separate, high-resolution files.

We look forward to receiving your revised submission.

Yours sincerely,

Richard Carson
Senior Editor
The Journal of Physiology

EDITOR COMMENTS

Reviewing Editor:

Ethics Concerns:

Please state in the methods section each time the individual mice are anaesthetised and for what purpose. Terminal procedures should also be explicitly stated.

Comments for Authors to ensure the paper complies with the Statistics Policy:

Please show individual data in the graphs where possible. On a couple of occasions a

Effect sizes may be a more important indicator of reliability in this type of study.

Comments to the Author:

Please clarify some of the missing details in the methods regarding anaesthesia and the method of terminal euthanasia.

The study comprises a serious amount of data, showing some of these raw traces may be great in the figures.

Expanding the analysis to pay some attention to the somatotopy of the motor pool and the degree of synergy may be very informative.

Senior Editor:

Comments to the Author:

Please report the degrees of freedom (or sample size) associated with each statistical test for which a p value is reported.

In all cases in which a measure of dispersion is shown in figure (e.g., Figure 2 Panel D), the basis upon which the error bars were calculated should be specified in the figure legend.

More generally, greater details concerning the inferential tests that generated the "outcome" p values should be provided.

REFEREE COMMENTS

Referee #1:

This study uses a combination of optogenetic stimulation and EMG recordings to examine how networks of spinal

interneurons control coordinated muscle activity. The researchers found that specific interneurons could be linked to patterns of muscle activation (muscle synergies), confirming that these circuits play a key role in movement control. The researchers also observed that the way interneurons control muscle synergies can change depending on the experimental conditions, suggesting a high degree of flexibility in the spinal cord's control strategies.

- The experimental design is well-conceived and technically sound.
- The authors used optogenetics to selectively stimulate spinal interneurons and recorded both neural and EMG activity. This approach allowed them to investigate the role of spinal interneurons in generating hindlimb muscle coordination patterns. The use of CNTF electrodes is innovative.
- The authors used spike-triggered averaging of EMG and muscle field analysis to assess the functional connectivity between spinal interneurons and motoneurons. They also used non-negative matrix factorization to extract muscle synergies from the EMG data.
- The muscle fields of spinal interneurons were found to be highly similar to the muscle synergies extracted from EMG recordings. This finding is consistent with previous studies in frogs and monkeys.
- The muscle fields of the same interneurons changed across different stimulation conditions. This finding suggests that the spinal cord can flexibly adjust its output depending on the current state of the network. The authors propose a plausible mechanism to explain this state-dependent encoding. This raises important questions about the neural mechanisms underlying the selection and combination of muscle synergies.

Minor comments:

- The authors should discuss the limitations of using anesthetized animals, which could alter the activity of spinal circuits and influence the results. It would be helpful to acknowledge this limitation.
- The authors should provide more details about the criteria used to classify spinal interneurons. The classification based on onset latency and peak width of SpTA is briefly described, but it would be helpful to provide more information about the rationale for these criteria.
- The authors should clarify the relationship between the different forms of synergy modification and the different stimulation conditions. It would be helpful to more explicitly state which form of modification (merging, unmatched, and preserved) were observed under which stimulation conditions.

Referee #2:

The authors Borong HE et al submitted a manuscript entitled "State-Dependent Neural Representations of Muscle Synergies in the Spinal Cord Revealed by Optogenetic Stimulation." This manuscript offers additional evidence that hindlimb muscle synergies exist and that the impact of the spinal circuit synergies on motor output is stimulation/state-dependent. Overall, the manuscript was well written, and the results will be of interest to many in the motor circuit community. The experimental approach is both challenging and complicated which leads to a low yield of data and is an inherent weakness in these types of studies and the overall impact. That being said, the sample rate is higher in this manuscript than what has been presented in previous publications indicating an advancement in approaches with the novel costimulation and altered stimulation paradigms. Moreover, I generally agree with the overall conclusions but there could be additional technical and anatomical limitations contributing to the resolution of synergies (see #9 and #14 below). I appreciated the hypothetical models presented in Figure 7 and think this aided a great deal in the interpretation of results. I do not have several questions and requests that, in my humble opinion, will increase the readability of the manuscript.

In general, some of the figure legends were not sufficient at describing the details in the figure and placing them in context with the manuscript. I had to refer back to the methods and results often to understand the data presented in figures, and sometimes I was not able to interpret completely. In figure legends, please define all abbreviations used in the figures in addition to statistical symbols. Please define which statistical test was used in figure legends.

1. In the introduction, the following sentence "Premotor interneurons in the spinal ventral horn (at depth of about 700 μm from dorsal spinal surface) exhibited post-spike facilitations with significant effects on multiple muscles at the same time at a long latency, and these neurons are thought to be involved in neuronal representations of muscle synergies"- can you please state the species mentioned for the Hart and Gizter 2010 reference? I thought this was a frog prep, but seems to be

justification for the depth used in the mouse model. This is a little confusing. Since the neuronal population that is contributing to the synergies are unknown, can you provide some additional context as to how the location/depth of CNTF electrodes were selected?

2. I appreciated the authors description and justification of the use of CNTF and the fabrication of the CNTF. This was well described and will be helpful to other scientists.

3. In methods, in the CNTF electrode fabrication, some chemical names need to be formatted appropriately (subscripts).

4. In methods under "Spinal electrophysiology recording using CNTF electrodes". This part is not consistent with Figure 1. Specifically, "two CNTFs were typically inserted into L2 and/or L3 spinal segments" but the figure shows something different. Also, where the two electrodes places ipsilateral or bilateral to one another? Please clarify.

5. Methods 2.3 Spike Triggered Average of EMG and Muscle Fields. How is the raw EMG signal filtered? It should be somewhere in the range of 20- 500 hz to get entire raw EMG signal without causing aliasing (your low pass is good). The filters listed sound like secondary filters but I am unsure. Please clarify.

6. Figure 1- Please reference in text. This figure is critically important to convey the overall complexity of the experimental design. Please consider adding the optogenetic stimulation strategy and where stimulations are being performed (both single and double stims). This will help the reader have a better understanding of the experimental design.

7. In 3.1, please insert the phrase "recorded from" so that the sentence reads "in 13 units recorded from 9 mice,"

8. In 3.1- What is the reason for reporting significant differences between preferred synergies and non-preferred synergies?

9. In 3.1, in reference to the sentence "The unmatched muscle synergies may correspond to muscle fields of other neurons that were recorded in the experiments". Could it also be motor units firing further away from the EMG electrodes? Usually, the fields of differential amplifier EMG electrodes are small and will not capture the entire muscle. This can be included in discussion as well (section 4.1).

10. Figure 2A- please reference in the text.

11. Figure 2B- based on description in methods, why are there so many negative latencies? These latencies are not aligning with the "classification of spinal interneurons" section.

12. Figure 2C- What is W1, W2 and W3?

13. Figure 3. Not sure exactly how to read this. Are the red bars the spike triggered averages with Blue and Gray the EMG synergies?

14. Lumbrosacral motoneurone pools are organized medially to laterally, with medial pools innervating more proximal muscles and lateral pools innervating more distal muscles. I realize this is purely speculative, but do you think that the interneuron populations responsible for the synergies can also be organized in this fashion? Specifically, do you think that that a medial to lateral stimulation and/or recordings could produce different profiles and increase "preferred synergies"?

I believe upon some of the revisions that this study will advance the spinal motor circuit field.

END OF COMMENTS

Response to Editor Comments

EDITOR COMMENTS

Reviewing Editor:

Ethics Concerns:

Please state in the methods section each time the individual mice are anaesthetised and for what purpose. Terminal procedures should also be explicitly stated.

Response: We appreciate the editor's comment regarding the clarity of the anesthesia and terminal procedures in the text. We have added the following statements in the section "Preparation and Animal Surgeries" to explicitly describe the purpose of anesthesia and the terminal procedure:

1. "Before each session, the mouse was anesthetized first with isoflurane (1.5-2%; mixed with 0.8L/min oxygen), followed by an intraperitoneal injection of ketamine/xylazine (100/10mg/kg) before implantation of intramuscular EMG electrodes and laminectomy. On the first day of surgery, as part of post-operative care, buprenorphine (0.05mg/kg) was administered subcutaneously to alleviate pain." (line 163 to 167)
2. "At the end of the experiment, while still under deep anesthesia, the animal was perfused transcardially as described in the section "Histology" below. Throughout the entire experiment, each animal was anesthetized once before the terminal procedure." (line 170 to 173)

Comments for Authors to ensure the paper complies with the Statistics Policy:

Please show individual data in the graphs where possible. On a couple of occasions a .

REVISION CHECKLIST:

We look forward to receiving your revised submission.

Yours sincerely,

Richard Carson
Senior Editor
The Journal of Physiology

REQUIRED ITEMS

- The Journal of Physiology funds authors of provisionally accepted papers to use the premium BioRender site to create high resolution schematic figures. Follow this link and enter your details and the manuscript number to create and download figures. Upload these as the figure files for your revised submission. If you choose not to take up this offer, we require figures to be of similar quality and resolution. If you are opting out of this service to authors, state this in the Comments section on the Detailed Information page of the submission form. The link provided should only be used for the purposes of this submission. Authors will be charged for figures created on this premium BioRender account if they are not related to this manuscript submission.

EDITOR COMMENTS

Reviewing Editor:
We appreciate the responses to the editorial and reviewer queries.

Some discussion points need to be expanded upon, in particular the neuroanatomical relationships that relate to the present findings.

REFEREE COMMENTS

Referee #1:

I am satisfied with the answers to my comments as well as those to my fellow reviewer.

Referee #2:

The authors Borong HE et al submitted a revised manuscript entitled "State-Dependent Neural Representations of Muscle Synergies in the Spinal Cord Revealed by Optogenetic Stimulation." This revised manuscript offers additional evidence that hindlimb muscle synergies exist and that the impact of the spinal circuit synergies on motor output is stimulation/state-dependent. This manuscript was well written, and the results will be of interest to many in the motor circuit community. The authors addressed all the concerns from the first review thoughtfully. The figure legends now sufficiently describe and clarify the figures helping the reader to understand their complicated data sets. Some of the revisions in the figures help clarify their approach and interpretation of their data sets. I particularly appreciate the addition of individual data points in Fig 2. Overall, I am satisfied with the revisions, and I believe these revisions will help improve the overall impact of the work.

There is one area I would like the authors to consider. Two reviewers asked questions about somatotopic organization of the motor pools and synergies, albeit they were different questions. This is an area that is of interest in the motor circuit field and something that kept popping in my mind as I was reading the manuscript. I would like to encourage the authors to consider adding a few discussion sentences speculating on spatial organization of interneurons and/or synergies relative to motor pool organization. The authors responses to the reviewers were well reasoned and thought-provoking and I think that some of their thoughts, as discussed with the reviewers, will be of interest to the wider community. In my opinion, this study is profoundly thought-provoking and can inspire new lines of research and some of this discussion can foreshadow that and even pave the way.

END OF COMMENTS

The authors Borong HE et al submitted a revised manuscript entitled “State-Dependent Neural Representations of Muscle Synergies in the Spinal Cord Revealed by Optogenetic Stimulation.” This revised manuscript offers additional evidence that hindlimb muscle synergies exist and that the impact of the spinal circuit synergies on motor output is stimulation/state-dependent. This manuscript was well written, and the results will be of interest to many in the motor circuit community. The authors addressed all the concerns from the first review thoughtfully. The figure legends now sufficiently describe and clarify the figures helping the reader to understand their complicated data sets. I particularly appreciate the addition of individual data points in Fig 2. Overall, I am satisfied with the revisions and I believe these revisions will help improve the overall impact of the work.

There is one area I would like the authors to consider. Two reviewers asked questions about somatotopic organization of the motor pools and synergies, albeit they were different questions. This is an area that is of interest in the motor circuit field and something that kept popping in my mind as I was reading the manuscript. I would like to encourage the authors to consider adding a few discussion sentences speculating on spatial organization of interneurons and/or synergies relative to motor pool organization. The authors responses to the reviewers were well reasoned and thought-provoking and I think that some of their thoughts, as discussed with the reviewers, will be of interest to the wider community. In my opinion, this study is profoundly thought-provoking and can inspire new lines of research and some of this discussion can foreshadow that and even pave the way.

Response to Editor Comments

Formatted: Centered

EDITOR COMMENTS

Formatted: Font: 12 pt

Reviewing Editor:

We appreciate the responses to the editorial and reviewer queries.

Some discussion points need to be expanded upon, in particular the neuroanatomical relationships that relate to the present findings.

We sincerely appreciate the thoughtful feedback from the Editor and Reviewers on our manuscript and we are grateful for the opportunity to improve the manuscript and believe these revisions have strengthened the work. We have expanded the Discussion section to further clarify the neuroanatomical relationships underlying our findings (see Discussion 4.1, line 637 to 662).

Formatted: Font: 12 pt

REFeree COMMENTS

Referee #1:

I am satisfied with the answers to my comments as well as those to my fellow reviewer.

We are pleased that our responses addressed the reviewer's questions satisfactorily.

Formatted: Font: 12 pt

Formatted: Font: 12 pt, Font color: Custom Color(RGB(4,50,255))

Referee #2:

Formatted: Font: 12 pt

The authors Borong HE et al submitted a revised manuscript entitled "State-Dependent Neural Representations of Muscle Synergies in the Spinal Cord Revealed by Optogenetic Stimulation." This revised manuscript offers additional evidence that hindlimb muscle synergies exist and that the impact of the spinal circuit synergies on motor output is stimulation/state-dependent. This manuscript was well written, and the results will be of interest to many in the motor circuit community. The authors addressed all the concerns from the first review thoughtfully. The figure legends now sufficiently describe and clarify the figures helping the reader to understand their complicated data sets. Some of the revisions in the figures help clarify their approach and interpretation of their data sets. I particularly appreciate the addition of individual data points in Fig 2. Overall, I am satisfied with the revisions, and I believe these revisions will help improve the overall impact of the work.

There is one area I would like the authors to consider. Two reviewers asked questions about somatotopic organization of the motor pools and synergies, albeit they were different questions. This is an area that is of interest in the motor circuit field and something that kept popping in my mind as I was reading the manuscript. I would like to encourage the authors to consider adding a few discussion sentences speculating on spatial organization of interneurons and/or synergies relative to motor pool organization. The authors responses to the reviewers were well reasoned and thought-provoking and I think that some of their thoughts, as discussed with the reviewers, will be of interest to the wider community. In my opinion, this study is profoundly thought-provoking and can inspire new lines of research and some of this discussion can foreshadow that and even pave the way.

Thank you for your suggestions and nice words. We agree that the spatial distribution of spinal interneurons and muscle synergies will be of interest to our community. We have added the following paragraphs to our Discussion 4.1 (line 637 to 662).

“While our study focuses on the functional encoding of muscle synergies in the spinal interneurons, the question of whether the neurons that encode the different synergies exhibit a topographical organization relative to that of the motoneuronal pools of the different muscles remains intriguing. Anatomical studies have established that the motoneuronal pools of the hindlimb muscles are organized as overlapping anteroposterior columns spanning different numbers of spinal segments (1). A recent work has shown that the premotor interneurons that activate motor synergies of the distal muscles (including the gastrocnemius) have a widespread distribution across multiple spinal segments (L2 to L5) and Rexed laminae, though they are concentrated in the deep medial dorsal horn (2). Consistent with this result, stimulation studies in the frog (3, 4) and mouse (5, 6) further demonstrated that the spinal loci whose focal stimulation recruited the same muscle synergy could have a very wide distribution that spans multiple spinal segments. The distributions of the different muscle synergies also overlap extensively. Together, these results suggest a complex spatial organization in which interneurons for the different muscle synergies are intermingled and distributed across segments, with unclear relationships with the motoneuronal distributions of the muscles coordinated by these synergies.

As our results argue, the encoding of muscle synergies likely depends not only on the connectivity between individual interneurons and the motoneurons, but also on their connectivity patterns with the sensory neurons and other premotor interneurons within the spinal circuits. This network-level complexity implies that the representation of synergies may be altered under different conditions. Likewise, the synergy topography that reflects the distributions of the PreM-INs and upstream interneurons capable of recruiting the muscle synergies may also adapt when the spinal cord is subjected to different afferent and descending stimulations. Systematic mapping of interneuron populations combined with connectivity analysis will be critical to resolve whether the spinal topography of the synergy-encoding interneurons follow a more-or-less invariant map, or emerge dynamically from the interactions between neurons in the spinal network and network inputs.”

1. **Mohan R, Tosolini AP, Morris R.** Targeting the motor end plates in the mouse hindlimb gives access to a greater number of spinal cord motor neurons: An approach to maximize retrograde transport. *Neuroscience* 274: 318–330, 2014. doi: 10.1016/j.neuroscience.2014.05.045.
2. **Levine AJ, Hinckley CA, Hilde KL, Driscoll SP, Poon TH, Montgomery JM, Pfaff SL.** Identification of a cellular node for motor control pathways. *Nat Neurosci* 17: 586–593, 2014. doi: 10.1038/nn.3675.
3. **Saltiel P, Wyler-Duda K, d’Avella A, Ajemian RJ, Bizzi E.** Localization and Connectivity in Spinal Interneuronal Networks: The Adduction–Caudal Extension–Flexion Rhythm in the Frog. *Journal of Neurophysiology* 94: 2120–2138, 2005. doi: 10.1152/jn.00117.2005.
4. **Saltiel P, d’Avella A, Wyler-Duda K, Bizzi E.** Synergy temporal sequences and topography in the spinal cord: evidence for a traveling wave in frog locomotion. *Brain Struct Funct* 221: 3869–3890, 2016. doi: 10.1007/s00429-015-1133-5.

Formatted: Font color: Custom Color(4,50,255)

Formatted: Font color: Custom Color(4,50,255)

Formatted: Font color: Custom Color(4,50,255)

Formatted: Font color: Custom Color(4,50,255)

Formatted: Font color: Custom Color(4,50,255)

Formatted: Font color: Custom Color(4,50,255)

Formatted: Font color: Custom Color(4,50,255)

Formatted: Font color: Custom Color(4,50,255)

Formatted: Font color: Custom Color(4,50,255)

Formatted: Font color: Custom Color(4,50,255)

5. **Caggiano V, Cheung VCK, Bizzi E.** An Optogenetic Demonstration of Motor Modularity in the Mammalian Spinal Cord. *Sci Rep* 6: 35185, 2016. doi: 10.1038/srep35185.
6. **Salmas P, Cheung VC.** Gradient descent decomposition of force-field motor primitives optogenetically elicited for motor mapping of the murine lumbosacral spinal cord. *Zoological Research* 44: 604, 2023.

Dear Professor Cheung,

Re: JP-RP-2025-288073R2 "State-Dependent Neural Representations of Muscle Synergies in the Spinal Cord Revealed by Optogenetic Stimulation" by Borong HE, Paola SALMAS, Jing ZHANG, Xiaojie DUAN, and Vincent C. K. Cheung

We are pleased to tell you that your paper has been accepted for publication in The Journal of Physiology.

Yours sincerely,

Richard Carson
Senior Editor
The Journal of Physiology

If you would like to receive our 'Research Roundup', a monthly newsletter highlighting the cutting-edge research published in The Physiological Society's family of journals (The Journal of Physiology, Experimental Physiology, Physiological Reports, The Journal of Nutritional Physiology and The Journal of Precision Medicine: Health and Disease), please click this link, fill in your name and email address and select 'Research Roundup':
<https://www.physoc.org/journals-and-media/membernews>

- You can help your research get the attention it deserves! Check out Wiley's free Promotion Guide for best-practice recommendations for promoting your work at: www.wileyauthors.com/eeo/guide. You can learn more about Wiley Editing Services which offers professional video, design, and writing services to create shareable video abstracts, infographics, conference posters, lay summaries, and research news stories for your research at: www.wileyauthors.com/eeo/promotion.

EDITOR COMMENTS

Reviewing Editor:

I thank the authors for their response to Reviewer #2.